# The Interplay between Aqueous Replacement Reaction and the Phase State of Internally Mixed Organic/ammonium Aerosols

**Hui Yang, Fengfeng Dong, Li Xia, Qishen Huang, Shufeng Pang, Yunhong Zhang**

School of Chemistry and Chemical Engineering, Beijing Institute of Technology, Beijing, 100081, People's Republic of China

**Correspondence**: Qishen Huang (qishenh@bit.edu.cn), Shufeng Pang (sfpang@bit.edu.cn)

**Abstract.** Atmospheric secondary aerosols are often internally mixed with organic and inorganic components, particularly dicarboxylic acids, ammonium, sulfate, nitrate, and chloride. These complex compositions enable aqueous reaction between organic and inorganic species, significantly complicating aerosol phase behavior during aging and making phase predictions challenging. We investigated carboxylate/ammonium salt mixtures using attenuated total reflection flourier transformed infrared spectroscopy (ATR-FTIR). The mono-, di- and tri-carboxylates included sodium pyruvate (SP), sodium tartrate (ST), and sodium citrate (SC), while the ammonium salts included $NH_4NO_3$, $NH_4Cl$, and $(NH_4)_2SO_4$. Our results demonstrated that aqueous replacement reactions between carboxylates and ammonium salts was promoted by the formation and depletion of $NH_3$ as relative humidity (RH) changed. For SP/ammonium aerosols, $NaNO_3$ and $Na_2SO_4$ crystallized from 35.7% to 12.7%, and from 65.7% to 60.1% RH, respectively, lower than pure inorganics ($62.5\pm9-32$% RH for $NaNO_3$ and $82\pm7-68\pm5$% RH for $Na_2SO_4$). Upon hydration, the crystalline $Na_2SO_4$ and $NaNO_3$ deliquesced at 88.8%–95.2% and 76.5$\pm$2–81.9%, higher than those of pure $Na_2SO_4$ (74$\pm$4%–98% RH) and $NaNO_3$ (65–77.1$\pm$3% RH). In contrast, reaction between ST or SC and $(NH_4)_2SO_4$ was implete due to the gel structure at low RH. Unexpectedly, aqueous $Na_2SO_4$ crystallized upon humidification in ST/$(NH_4)_2SO_4$ particles at 43.6% RH and then deliquesced with increasing RH. This is attributed to increased ion mobility in viscous particles, leading to nucleation and growth of $Na_2SO_4$ crystal. Our findings highlight the intricate interplay between chemical components within organic/inorganic aerosol, the impact of replacement reactions on aerosol aging and phase state, and subsequently on atmospheric processes.

**1 Introduction**

The phase state of atmospheric aerosols is arguably one the most crucial properties (Freedman et al., 2024) which dictates air quality, climate and human health via processes including light absorption and scattering (Nemesure et al., 1995), cloud condensation nuclei (CCN) activity (Kreidenweis et al., 2005), chemical diffusion in aerosols, gas-particle partitioning (Shiraiwa et al., 2012), and heterogeneous reactions (Liu et al., 2016). However, the complex aerosol composition and various reaction pathways, results in complicated aerosol phase behavior that is hard to predict (Zhang et al. 2024). Understanding the phase behaviors and water content of atmospheric aerosols is therefore essential for elucidating their impacts on global climate and tropospheric chemistry.

Atmospheric aerosols are often composed of inorganic sulfates, nitrates, and organic species in varying mixing ratios (Riemer et al., 2018; Huang et al., 2014; Trebs et al., 2005; Zhou et al., 2019). The interaction between organic and inorganic components can alter aerosol phase state and hygroscopicity (Shi et al., 2017; Wang et al., 2017; Marcolli et al., 2006). Previous works have investigated the effect of organic acids on inorganic salts, showing the impact of chemical composition and organic mass fractions (Huang et al., 2022; Jing et al., 2016). Recent field observations highlighted the significant role of organic acids in modifying the phase behavior of ammonium sulfate aerosols, further complicating the prediction of aerosol phase states (Li et al., 2021; Zhang et al., 2022; Kirpes et al., 2022). For atmospheric aerosols undergoing liquid–liquid phase separation (LLPS) with an organic shell and an inorganic core, balck carbon redistribution can be induced, which leads to a more compact morphology and a reduction in the absorption enhancement effect by 28 %–34% (Zhang et al. 2022). In addition, Organic components can decrease aerosol surface activity (Petters et al., 2007), and drive the formation of viscous states at low humidity, leading to limited water absorption (Reid et al., 2018 ). Zhu et al (2022) found that aqueous NaCl and $(NH_4)_2SO_4$ effloresced during humidification when they mixed with ultraviscous gluconic acid, showing high viscosity is another key regulator.

Aerosol phase state significantly influences chemical reaction within aerosols (Herrmann et al., 2015). Phase changes of aerosol components can enhance chemical reactions (Ma et al., 2022; Shao et al., 2018; Chen et al., 2021). For example, the photochemical HONO production from internally mixed $NaNO_3$-dicarboxylic acid aerosols is governed by the phase state of dicarboxylic acids (Li et al., 2024). In aqueous aerosols, the ions between two ionic compounds can exchange to form new compounds through aqueous replacement reactions. For instance, when oxalic acid mixes with nitrates, the $H^+$ ions can combine with $NO_3^-$ ions to form $HNO_3$, which off-gases,

leading to the formation of metal oxalate salts in the aerosol (Ma et al. 2019). Wang et al (2019) first reported the aqueous replacement reaction between organic salts and ammonium sulfate to form organic acid, accompanying solid $Na_2SO_4$ foramtion. These reactions can generate gaseous compounds or less hygroscopic compounds, causing the substitution of weak bases for strong bases in aerosols (Yang et al 2019; Du et al 2020; 2021; Chen et al, 2022).

Despite the critical role of aerosol phase state, our understanding of the interplay between aerosol component reactivity, phase behavior and water content remains limited. Reactions, such as replacement reaction, can alter aerosol phase state. The complexity of atmospheric aerosol phase states - including the coexistence of crystalline solid and liquid (Yang et al., 2019), liquid-liquid phase separation (Zhou et al., 2019), and solid in a viscous phase

(Zhu et al., 2022) - can lead to inhomogeneity of aerosol components, which in turn affects their reactivity. (Zong et al., 2022).The crystallization that accompanies these chemical reactions is a multi-step process involving ion migration, formation of ion pairs, nucleation, and growth processes. The extent of reaction, as well as the efflorescence relative humidity (ERH) and deliquescence relative humidity (DRH), can vary depending on the nature and molecular structure of the compounds involved. Therefore, investigating the interplay between

different organic and inorganic salts in terms of their reactivity and aerosol phase state is crucial for understanding atmospheric aerosols.

Organic acids are key components of atmospheric aerosols, each containing various functional groups with distinct physicochemical properties. Similarly, sulfates, chlorides, and nitrates exhibit unique characteristics. To investigate the impact of organic salts on phase behavior and chemical reactions, we mixed sodium tartrate,

sodium citrate, and sodium pyruvate—representing mono-, di-, and tri-carboxylates—with ammonium sulfate. We also examined the effects of inorganic ions by mixing ammonium nitrate with sodium pyruvate. Molar ratios of 1:2, 1:1, and 2:1 were used to simulate surplus organic salts, equal, and excessive inorganic salts in mixed aerosols, reflecting variations by location, aerosol source, and season. Attenuated total reflection Fourier-transform infrared spectroscopy (ATR-FTIR) was employed to monitor changes in phase and composition during

chemical reactions. This study improves our understanding of the interplay between aerosol composition, replacement reactions, and phase states in organic-inorganic aerosols, which is crucial for better predicting aerosol phase states and assessing their climate effects.

## 2 Materials and methods

### 2.1 Sample preparation

The 0.2 mol·L$^{-1}$ mixture solutions of organic acid salts and ammonium salts were prepared using triple distilled water. The organic acid salts include sodium pyruvate (SP, Aladdin Reagent Co., Ltd., ≥99.0%), sodium citrate (SC, Beijing Chemical Reagents Company, ≥99.0%), and sodium tartrate (ST, Beijing Chemical Reagents Company, ≥99.0%). The ammonium salts include $(NH_4)_2SO_4$, $NH_4NO_3$ and $NH_4Cl$ (all from Beijing Chemical Reagents Company, ≥99.0%). In a mixture system, one organic acid salt was mixed with one type of ammonium salts at various molar ratios (e.g., 2:1, 1:1, 2:3, and 1:2). In the atmosphere, the organic to inorganic ratio varies regionally, but mostly remains at the same order of magnitude. Thus, the selected ratios herein can effectively represent different regions. The mixture solutions were aerosolized into aerosol droplets with an average diameter of 5μm. All compounds were used without further purification before experiment. The molecular structure and properties of the compounds involved in this study are listed in Table 1.

**Table 1: The molecular structure of inorganic and organic compounds**

| Compound | Molecular structure | Dissociation constant of conjugate acid | Solubility in 100 g H$_2$O (20°C) | Molecular weight (Da) |
|---|---|---|---|---|
| Ammonium nitrate | $NH_4NO_3$ | NA | 190 g | 80.043 |
| Ammonium chloride | $NH_4Cl$ | NA | 37.2 g | 53.49 |
| Ammonium sulfate | $(NH_4)_2SO_4$ | NA | 75.4 g | 132.14 |
| Sodium pyruvate | | $K_a = 3.2 \times 10^{-3}$ | 47 g | 82.03 |
| Sodium citrate | | $K_{a,1} = 7.4 \times 10^{-4}$ $K_{a,2} = 1.7 \times 10^{-5}$ $K_{a,3} = 4.0 \times 10^{-7}$ | 154 g | 258.07 |
| Sodium tartrate | | $K_{a,1} = 1.04 \times 10^{-3}$ $K_{a,2} = 4.55 \times 10^{-5}$ | 33.3 g | 194.05 |

### 2.2 ATR-FTIR measurement

The ATR-FTIR measurement was conducted using a FTIR spectrometer (Nicolet Magna-IR model 560) equipped with a liquid-nitrogen-cooled mercury-cadmium-telluride (MCT) detector. A RH-controlling system, composed of a high-purity water reservoir, mass flowmeter and vacuum pump, was connected to the FTIR spectrometer.

The RH was controlled by the mixing ratio of a wet nitrogen gas flow (saturated by water vapor), and a dry nitrogen gas flow. The total flow rate of the two nitrogen gas flows was 500 mL min$^{-1}$, and the flow rate of the wet and dry nitrogen gas flow were controlled by mass flow controllers. During the experiments, the RH was monitored by humidity with a precision of 0.5% and an accuracy of 1% and temperature sensors coupled to a data logger at the outlet of the sample chamber. The detailed instrumental information can be found in our study (Zhang et al., 2014). Prior to the ATR-FTIR measurement, the RH was kept at the highest level in the sample chamber. The aerosols were atomized and introduced into the sample chamber to be deposited onto the ZnSe substrate under the high RH condition. At room temperature, the spectral resolution and the spectral range of the ATR-FTIR are 4 cm$^{-1}$ and 800-4000 cm$^{-1}$, respectively.

**2.3 Data processing**

The IR spectra of aerosols were obtained by subtracting IR of water vapor at corresponding RH from raw IR spectra without any smoothening. The water content at each RH level was gained by integrating the bands at 3360-3690 cm$^{-1}$ in the corresponding IR spectra. The relative water content was achieved by normalizing the peak area of the water band (3360-3690 cm$^{-1}$) at a specific RH to that at the maximum RH.

**3 Results and Discussion**

The phase behaviour of mixed aerosols is dependent upon the chemical composition, molar ratio and chemical process. Herein we selected three organic salts including SP, ST and SC, along with NH$_4$Cl, NH$_4$NO$_3$ and (NH$_4$)$_2$SO$_4$, to study the interplay between phase state, composition evolution, and aqueous phase replacement reactions in aerosols during RH changing cycles. This section consists of seven parts. In *part 3.1*, we analysed the IR spectra of pure organic salts and inorganic salts to obtain the differences in characteristic peaks for aqueous and solid phases. Based on this, in *part 3.2*, SP was mixed with different ammonium salts with various stoichiometric ratio to form aerosols, and the infrared spectra of the mixed aerosols were tested to analyse the characteristic absorption peaks in the second section. The results showed that SP underwent a substitution reaction with the ammonium salts, and the mixed aerosols containing different ammonium salts exhibited distinct phase behaviours. Therefore, in *parts 3.3 and 3.4*, we discussed in detail the substitution reactions and the resulting changes in water content and phase transition regions of the compounds. The phase transition regions of the reaction products were compared with those of the pure components. The relative content of the mixed components may vary in different regions. In *part 3.5*, we took the SP/NH4Cl system as an example to study the

impact of different component contents on phase transition behaviour. After investigating the hygroscopic behaviour of mixed aerosols containing different ammonium salts with the same organic salt, in *part 3.6*, we examined the phase evolution of aerosols induced by mixing different organic salts, SC and ST, with ammonium sulphate. By combining infrared spectroscopy and optical images, we discovered that the mixture of tartaric acid and ammonium sulfate not only underwent substitution reactions but also exhibited unexpected hygroscopic weathering behaviour. Therefore, in *part 3.7*, we discussed the causes of this special phase behaviour.

**3.1 Spectral change and phase behavior of pure inorganic and organic aerosols**

The IR spectra of individual components, namely, ammonium nitrate, ammonium sulfate, ammonium chloride, sodium pyruvate, sodium citrate, and sodium tartrate are shown in the Supporting Information (Fig. S1) and band assignments can be found in Table S1. The features of the IR spectra can be used to differentiate the phase state of each compound. For example, for sodium tartrate, the characteristic peaks of the aqueous phase and crystalline solid phase are located at 1069 cm$^{-1}$ and 1055 cm$^{-1}$, respectively. Sodium citrate exhibits double sharp peaks at 1308 and 1278 cm$^{-1}$ during crystallization. For pure sodium pyruvate, the characteristic peaks of solid phase appear at 1405 cm$^{-1}$, and the band at 1176 cm$^{-1}$ is shifted to 1186 cm$^{-1}$. As for crystalline solid NH$_4$Cl, bands at 3130 and 1402 cm$^{-1}$ are the characteristic IR peaks. When NH$_4$Cl was mixed with sodium pyruvate, the band at 1402 cm$^{-1}$ may overlap with the 1405 cm$^{-1}$ and become indistinguishable; therefore, the band at 3130 cm$^{-1}$ was used as the feature for crystalline solid NH$_4$Cl, and the band 1186 cm$^{-1}$ was applied to characterize crystalline solid sodium pyruvate. For NH$_4$NO$_3$ and (NH$_4$)$_2$SO$_4$, when the phase transition from aqueous to crystalline solid phase occurs, the NH$_4^+$ bands exhibit dissimilarity: shifting from 1448 cm$^{-1}$ to 1417 cm$^{-1}$ for NH$_4$NO$_3$, and from 1443 cm$^{-1}$ to 1412 cm$^{-1}$ for (NH$_4$)$_2$SO$_4$. When NH$_4$NO$_3$ and (NH$_4$)$_2$SO$_4$ are mixed with organic salts, the NH$_4^+$ bands can overlap with the characteristic peaks of sodium citrate and sodium tartrate in the same wavenumber region, leading to difficulties in assigning the peaks to specific crystalline solid ammonium salts.

The IR spectra on dehydration and hygroscopic behavior of organics (sodium pyruvate, sodium citrate, and sodium tartrate) during RH change were shown in Fig. 1. For pure organic compounds, the response of water content in aerosols can be applied to elucidate phase transition. Sodium pyruvate aerosols effloresced at 63-60.5% RH upon dehydration, corresponding to a sudden change in water content. In comparison, the water content of pure sodium citrate and sodium tartrate aerosols decreased gradually upon dehumidification, and the retained normal water content of 0.25 and 0.15 for ST and SC, respectively, at the lowest RH, indicating the formation of a viscous state instead of crystalline solid phase (Mikhailov et al., 2009). The formation of viscous phase was also

consistent with the absence of crystalline solid features in IR spectra and the residual water peak at 3360 cm$^{-1}$.

Upon humidification, sudden water uptake occurred at 47.1–49.8% RH for sodium citrate, and at 55.7–59% RH

for sodium tartrate aerosols, where an error margin is 1%. Previous study has shown that similar abrupt water

absorbance during humidification was observed for viscous sucrose (Madawala et al., 2021) and MgSO$_4$ aerosols

(Wang et al., 2018), which was attributed to the hydration of aerosols during the phase transition from gel state to

aqueous state. Hence, the abrupt change in water content of the sodium citrate and sodium tartrate aerosols is due

to the phase transition from a viscous state to an aqueous state.

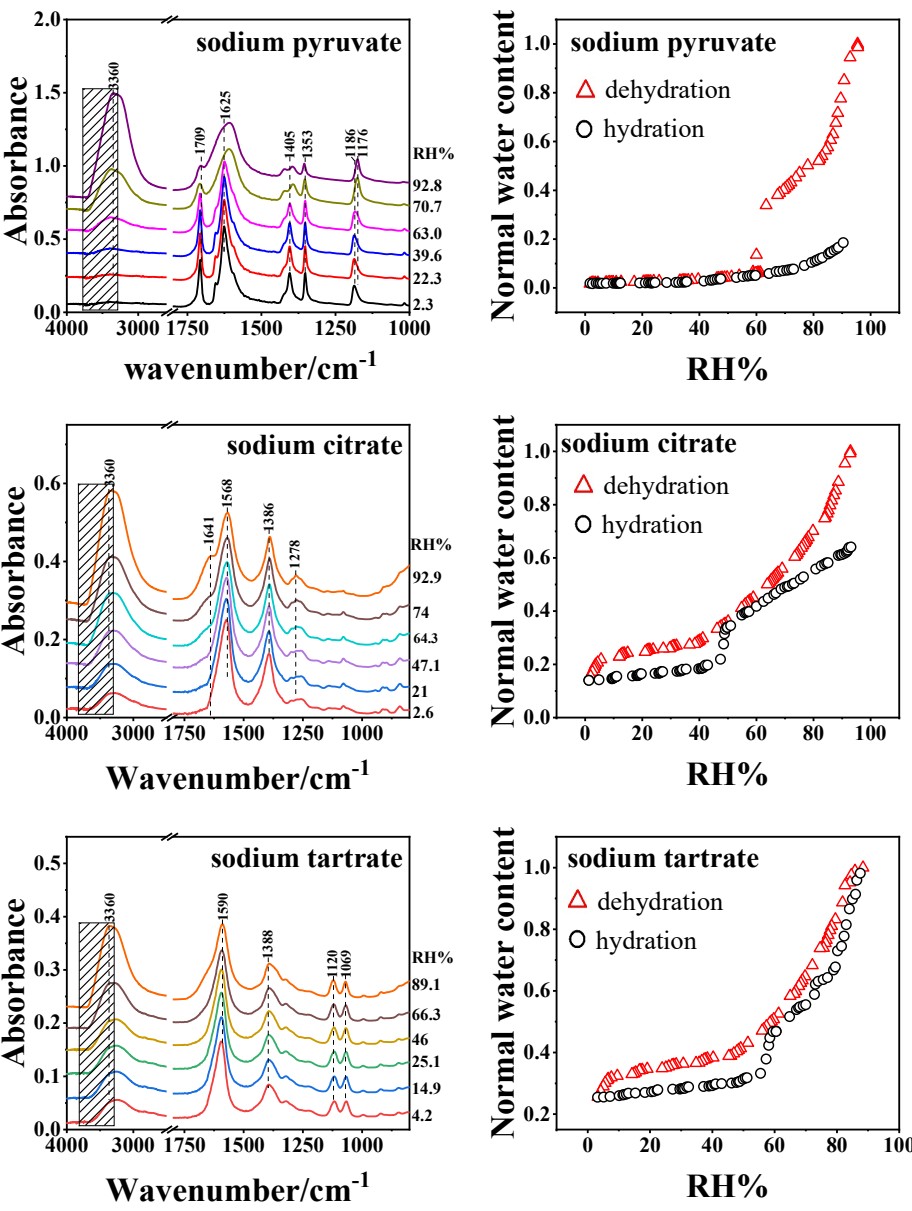

**Figure 1: The IR spectra of organic salts on dehydration and hygroscopic behavior during a down-up RH cycle. The
shaded area shows the chosen integration region for liquid water. The spectra for sodium pyruvate was previous**

**reported by Yang et al (2019).**

### 3.2 Spectral change and phase behavior of mixture organic/inorganic aerosols

Sodium pyruvate (SP) was mixed with various ammonium salts, including $(NH_4)_2SO_4$, $NH_4Cl$ and $NH_4NO_3$ as a molar ratio of 2:1, 1:1 and 1:2, respectively. Fig. 2 illustrates their IR spectra as the RH decreased. The data for $SP/(NH_4)_2SO_4$ (Fig. 2(a)) have been reported by Yang et al. (2019). The band at 3400 $cm^{-1}$, assigned to OH stretching mode of water, became weaker upon dehydration, suggesting a gradual loss of liquid water content. Concurrently, the bands at 1708 $cm^{-1}$ and 1355 $cm^{-1}$ gradually diminished, indicating a decrease in the quantity of SP in the aerosols phase. In the blue frame of Fig. 2(a), the enlarged IR feature of SP depicted the gradual weakening of 1176 $cm^{-1}$ peak, which disappeared at 65.7% RH, implying the complete SP depletion from the aerosols. The overlapping nature of the $\nu_4$-$NH_4^+$ mode, located at ~1440 $cm^{-1}$ in the aqueous phase or 1412 $cm^{-1}$ in the crystalline solid phase, with the bands of SP at 1424 $cm^{-1}$ or 1406 $cm^{-1}$, renders the determination of the phase state of $(NH_4)_2SO_4$ ambiguous. Accompanying the SP depletion, we noted the appearance of a band at 1132 $cm^{-1}$ at 65.7%, followed by its subsequent increase with further decreasing RH. Tan et al (2014) investigated the efflorescence of $Na_2SO_4$, and assigned the 1132 $cm^{-1}$ band to crystalline solid $Na_2SO_4$. In addition, the $\nu_1$-$SO_4^{2-}$ mode effectively discerns the sulfate phase state, with a peak at 980 $cm^{-1}$ indicating the aqueous state and a peak at 996 $cm^{-1}$ denoting the crystalline solid state (Miñambres et al., 2013). The purple frame of Fig. 2(a) showed the emergence of the 995 $cm^{-1}$ band at 65.5% RH, accompanied by the diminishing 980 $cm^{-1}$ band. At 60.1% RH, the 980 $cm^{-1}$ disappeared, indicating the complete crystallization of $Na_2SO_4$.

For $SP/NH_4Cl$ aerosols, the IR bands for SP at 1708 $cm^{-1}$ 1355 $cm^{-1}$ and 1627 $cm^{-1}$ became weaker at lower RH levels (Fig. 1(b)), indicating the decrease in SP content. As depicted in the blue frame of Fig. 2(b), both the crystalline solid IR band (1186 $cm^{-1}$) and the aqueous IR band of SP (1176 $cm^{-1}$) can be found at and below 23.7% RH, implying the coexistence of liquid and crystalline solid SP in $SP/NH_4Cl$ aerosols. Additionally, the band at 1404 $cm^{-1}$ became prominent at 42.5% RH. Given the comparable peak height between 1353 $cm^{-1}$ and 1405 $cm^{-1}$ in the IR spectrum of crystalline solid SP, the band at 1404 $cm^{-1}$ likely originated from crystalline solid $NH_4Cl$, consistent with the sharp bands at 3130 and 3036 $cm^{-1}$ (Max et al., 2013). According to previous studies, NaCl should form in the system, however, the formation of NaCl cannot be confirmed merely from the IR spectrum as NaCl exhibits no IR absorptions.

For $SP/NH_4NO_3$ particles, we observed similar SP loss, denoted by weaker SP band adsorption upon dehydration, along with a stronger band at 1355 $cm^{-1}$, an IR feature corresponding to the efflorescence of $NaNO_3$

(1352 cm$^{-1}$ and 836 cm$^{-1}$) (Ren et al., 2016). When RH decreased to 35.7%, the band for crystalline solid NaNO$_3$ at 836 cm$^{-1}$ appeared (Fig. 2(c), purple frame), coexisting with the 829 cm$^{-1}$ band for aqueous NO$_3^-$, indicating the presence of liquid NO$_3^-$ across the RH range of 35.7%-32%. At 12.7% RH, only the band at 836 cm$^{-1}$ was present, suggesting the crystallized NaNO$_3$ without liquid NO$_3^-$ (Zhang et al., 2014). In the blue frame of Fig. 2(c), the band at 1176 cm$^{-1}$ shifts slightly to 1180 cm$^{-1}$ instead of the crystalline solid SP band at 1186 cm$^{-1}$, suggesting semi-solid SP rather than crystalline solid SP.

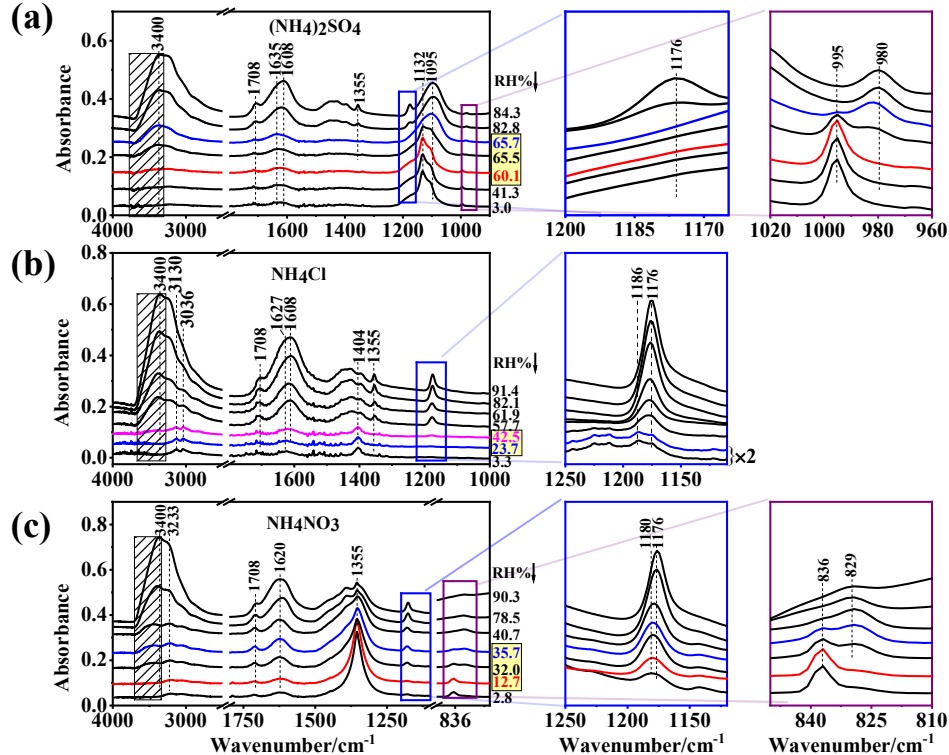

**Figure 2: The FTIR spectra of mixed aerosols containing sodium pyruvate and various ammonium salts measured in this study and in previous work. (a) ammonium sulfate (Yang et al., 2019), (b) ammonium chloride, (c) ammonium nitrate on dehydration. The spectral data for sodium pyruvate and ammonium sulfate was previously reported by Yang et al (2019). The shaded area shows the chosen integration region for liquid water. The yellow frame indicates the RH range when phase transition occurs (ERH of Na$_2$SO$_4$: 67.5-60.1%; onsets of ERH for SP and NH$_4$Cl: 23.7% and and 42,5%; ERH of NaNO$_3$: 35.7-12.7%).**

Following the dehydration, the hydration of three SP/ammonium aerosols was performed. Fig. S2 presents the IR spectra on hydration. As the RH increased, the water content of the three SP/ammonium mixture systems increased, characterized by the bands at 3355/3400 cm$^{-1}$ and 1640 cm$^{-1}$ as seen from Fig. S2. The characteristic bands of SP became negligible, indicating trace or no SP remaining in aerosols. While for SP/(NH$_4$)$_2$SO$_4$ aerosols, only the 995 cm$^{-1}$ band was found below 88.8% RH, indicating crystalline solid Na$_2$SO$_4$ without liquid SO$_4^{2-}$ in the particles. From 88.8% RH, two bands are present at 980 cm$^{-1}$ and 995 cm$^{-1}$, indicating the coexistence of liquid

$SO_4^{2-}$ and crystalline solid $Na_2SO_4$. As the RH continues to rise, the 995 cm$^{-1}$ band decreased rapidly, and then almost vanished at 95.2% RH, suggesting the deliquescence of aerosols. In the IR spectra of SP/NH$_4$Cl aerosols, the bands from crystalline solid NH$_4^+$, i.e. 3130 cm$^{-1}$, 3036 cm$^{-1}$ and 1404 cm$^{-1}$ decreased gradually and disappeared at 71.8% RH, indicating the depletion of ammonium during hydration. In the IR spectra of SP/NH$_4$NO$_3$ particles, the band at 829 cm$^{-1}$ begins to appear at 76.5%, indicating the dissolution of NaNO$_3$ solid.

When the RH reached 81.9%, the band at 836 cm$^{-1}$ disappeared, suggesting complete deliquescence of NaNO$_3$.

**3.3 The replacement reactions in SP/ammonium aerosols and the resulting hygroscopicity**

Above analysis on IR features showed the following reactions (1) – (3) in mixed SP/ammonium aerosols:

$$CH_3COCOONa\,(aq) + (NH_4)_2SO_4\,(aq) \xrightarrow{\text{RH decreasing}} CH_3COCOOH\,(g) + NH_3\,(g) + Na_2SO_4\,(s) \quad (1)$$

$$CH_3COCOONa\,(aq) + NH_4Cl\,(aq) \xrightarrow{\text{RH decreasing}} CH_3COCOOH\,(g) + NH_3\,(g) + NaCl\,(s) \quad (2)$$

$$CH_3COCOONa\,(aq) + NH_4NO_3\,(aq) \xrightarrow{\text{RH decreasing}} CH_3COCOOH\,(g) + NH_3\,(g) + NaNO_3\,(s) \quad (3)$$

It can be seen that pyruvic acid and ammonia are formed and depleted from particles alongside the crystalline solid formation of various inorganic salts during dehydration, which proceeded the replacement reaction between SP and ammonium. Herein, aerosol particles act as micro-reactors, with their larger specific surface area than bulk solution facilitating similar processes. In fact, the replacement reactions in aerosols driven by gas released or

230 compound formation with lower solubility have been reported in previous work. For example, Wang et al. (2017) observed the reaction in aerosols composed of oxalic acid and ammonium sulfate, which was driven by formation of lower hygroscopic ammonium hydrogen oxalate (NH$_4$HC$_2$O$_4$) and ammonium hydrogen sulfate (NH$_4$HSO$_4$) during the dehydration process. While Wang et al. (2019) addressed the fact of the formation of crystalline solid Na$_2$SO$_4$ from (CH$_2$)$_n$(COONa)$_2$ (n = 1, 2)/(NH$_4$)$_2$SO$_4$ aerosols upon dehydration. Building upon their findings,

crystalline solid NaNO$_3$ and NaCl are also formed, as shown in equations (2) and (3).

The phase state of aerosol particles strongly depends on their water content, which in turn is influenced by changes in ambient RH (Yeung et al., 2010). Conventionally, soluble inorganic aerosol particles spontaneously absorb water to form solution droplets upon hydration, and these droplets recrystallize upon dehydration. Pure inorganic particles, such as NaCl and (NH$_4$)$_2$SO$_4$, often undergo prompt phase transitions reflected by sudden

water loss or uptake (Martin, 2000). Conversely, pure organic aerosols exhibit more diverse hygroscopic behaviors, forming either (poly)crystalline states or semisolids (Kuang et al., 2010). When mixtures of compounds are present in aerosols, as is ubiquitous in atmospheric aerosols and in PM$_{2.5}$, phase transitions may occur for all

components or only a portion of them. Even if all constituents effloresce, the critical RH values vary compared to pure compounds. Consequently, the water content may not be indicative of the efflorescence or deliquescence process. For aerosols composed of sodium pyruvate and ammonium, the response of water content to ambient RH is dependent upon water vapor equilibrium, replacement reaction, product depletion, and phase transitions. Therefore, changes in water content and IR spectra of aerosols need to be considered collaboratively to accurately understand the reactions and phase transitions of these aerosols.

Fig. 3 shows the water content evolution during a RH cycle and their comparison with the phase transition points of compounds. The RH changes stepwise and the rate is < 5% min$^{-1}$. And the stay time at each level is 30 minutes to allow for equilibration between particles and the surrounding RH. A significant deviation between the RH at which phase changes occur and the RH at which water content changes abruptly can be observed. We note that the SP/$(NH_4)_2SO_4$ data was previously reported in Yang et al (2019). The sensitive RH range for SP/$(NH_4)_2SO_4$, where abrupt water loss occurs, ranged from $66.5\pm1$ to $59.8\pm1\%$ RH, overlapping with the phase change RH of $Na_2SO_4$ shown by red frame during dehydration in Fig. 3(a). Conversely, the deliquescence range of $Na_2SO_4$ (pink frame) occurs at higher RH values than the abrupt water absorption range of $74.4\pm1-87.3\pm1\%$ RH during hydration in Fig. 3(a). In the case of SP/$NH_4Cl$, the coexistence of aqueous and crystalline solid SP (red frame) occurs at RH values much lower than the range of sudden water loss ($61.7\pm1-42.2\pm1\%$ RH) in Fig. 3(b). When the RH indreases, obvious water uptake takes place from $68.9\pm1\%$ RH to $80.9\pm1\%$ RH. In previous studies, $NH_4NO_3$ was often believed to form a viscous gel, leading to gradual dehydration with decreasing RH (Li et al., 2017). However, the produced $NaNO_3$ undergoes an aqueous-to-solid transformation within the RH range of $35.7\pm1\%-12.7\%\pm1\%$ shown in red frame of Fig. 3(c), while maintaining almost constant water content. During hydration, excessive water absorption occurs in the range of $72.7\%\pm1-82.1\pm1\%\%$ RH, covering the deliquescence RH of $NaNO_3$ (pink frame of Fig. 3(c)).

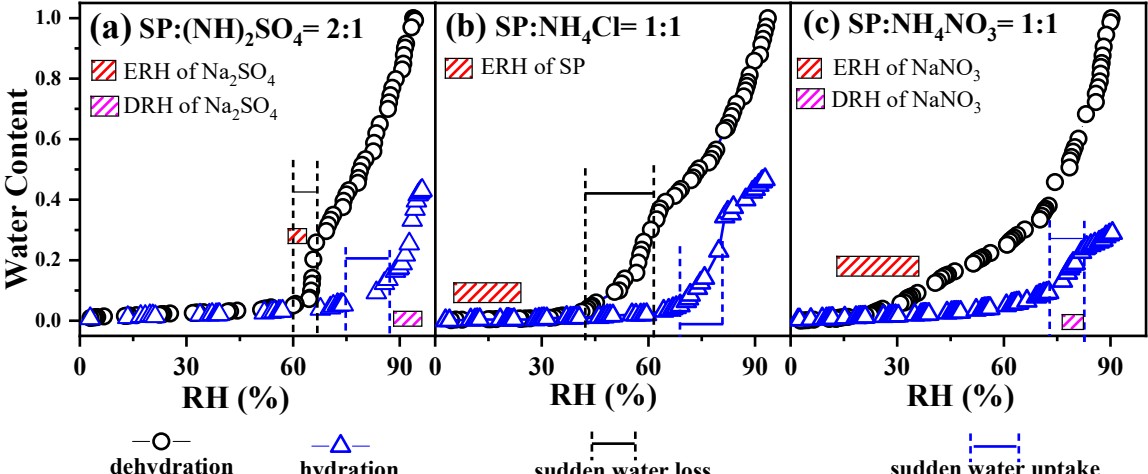

**Figure 3: Hygroscopicity curve of (a) SP: (NH₄)₂SO₄ =2:1 aerosols, (b) SP: NH4Cl =1:1 aerosols and (c) SP: NH4NO₃ =1:1 aerosols. The data for aerosols with SP: (NH₄)₂SO₄ =2:1 was previously reported by Yang et al (2019). The stoichiometric ratio between SP and (NH₄)₂SO₄ was set to 2:1 to keep the equivalent amount of organics and ammonium. In comparison, SP/NH4Cl aerosols and SP/NH4NO₃ aerosols also have equivalent amount of organics and ammonium.**

**3.4 The effect of organics on phase transition point**

When inorganic compounds are mixed with organics, the efflorescence and deliquescence points are modified due to intermolecular interactions. For instance, organic acids have been found to influence the phase transitions and water uptake of ammonium sulfate (AS) aerosols (Shi et al., 2017). Similarly, organic salts can enhance water uptake, sometimes reaching levels comparable to those of typical inorganic salts such as NaCl and (NH₄)₂SO₄.

Wu et al. (2011) observed a clear shift in the deliquescence relative humidity (DRH) of AS towards lower RH values for mixtures of AS with organic acid salts, resulting in enhanced water uptake relative to mixtures with organic acids alone. Additionally, Schroeder and Beyer (2016) noted that the onset DRH of organic salt/AS mixtures was consistently lower than that of the pure components, irrespective of the fraction of organic salts in the mixture.

Fig. 4 illustrates the RH ranges of efflorescence and deliquescence for comparison. It demonstrates the lower efflorescence relative humidity (ERH) of Na₂SO₄, NaNO₃, and SP in mixtures compared to those in pure aerosols, indicating that the presence of organics in the mixed particles can inhibit the crystallization of the produced inorganics. In contrast to crystallization, the DRH of Na₂SO₄ and NaNO₃ in mixtures were slightly higher than those of pure aerosols with a narrower DRH range. This deviates from previous experimental measurements and

thermodynamic model predictions, which suggested a significant reduction in aerosol DRH due to the mixing of

organic acids and inorganic salts (Bouzidi et al., 2020; Hodas et al., 2015).

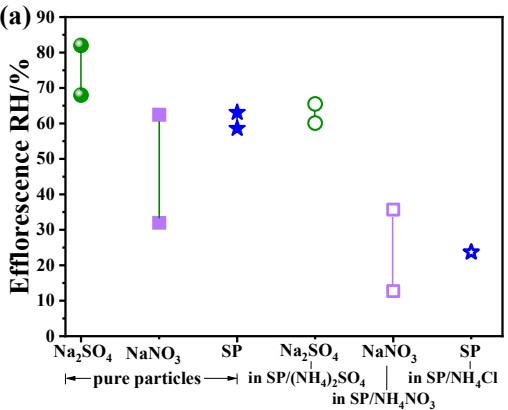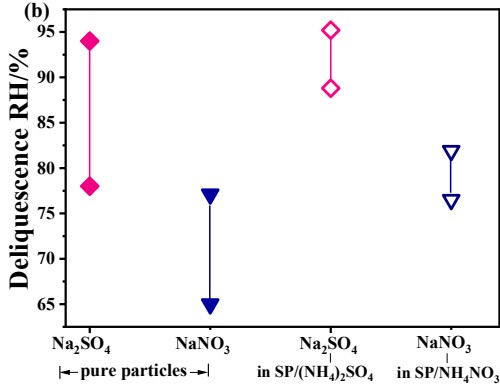

**Figure 4: The efflorescence RH (a) and deliquescence RH (b) for single component aerosols (pure particles) and SP/ammonium particles.**

**3.5 The effect of molar ratio on the replacement reaction and phase transition**

The chemical reactivity and phase state of aerosols are often influenced by individual compounds within internally mixed particles. In this study, SP/NH₄Cl particles were selected as a surrogate to investigate the dependence of component reconstruction and phase transition on the molar ratio of SP to NH₄Cl. Mixture SP/NH₄Cl aerosols were prepared with molar ratios of 1:2, 1:1, and 2:1. Fig. S3 presents the IR spectra in the range of 4000–800 cm⁻¹ for 2:1, 1:1, and 1:2 SP/NH₄Cl ratios. When the molar ratio of SP to NH₄Cl was 2:1, the band changes were reversible during dehydration and hydration cycle, with the absence of the 3130 cm⁻¹, $\nu$ (NH₄⁺) band of solid NH₄Cl. The band shift between 1608 cm⁻¹ and 1627 cm⁻¹ indicated the transition of SP between the liquid and crystalline solid phases.

For 1:1 SP/NH₄Cl particles, the IR spectra were shown in Fig.2(b) and Fig. S2(b) and described above. In the case of 1:2 SP/NH₄Cl aerosols, the band at 3130 cm⁻¹ was observed at 46.6% RH during dehumidification and disappeared at 77.7% RH during humidification. Following a dehumidification-humidification cycle, the overall quantity of compounds within the aerosol particles decreased, as reflected by the lower IR band intensity.

Fig. S4 displayed the IR spectra in the range of 1220-1120 cm⁻¹ during dehydration, providing detailed band shift and intensity information of the 1176 cm⁻¹ band. Additionally, the quantified peak position and integrated peak areas of SP/NH₄Cl particles, which can characterize the phase state and degree of chemical reaction, were depicted in Fig. 5. The band at 1176 cm⁻¹ indicates that the aerosols are in the aqueous phase, while its shift to 1186 cm⁻¹ signifies the transition to crystalline solid-phase particles. Thus, the onset of efflorescence relative humidity (ERH) can be identified by the appearance of the 1186 cm⁻¹ peak. From Fig. 5(a), the onsets of ERH

were approximately 61.7% and 23.7% for 2:1 and 1:1 SP/NH$_4$Cl particles, respectively. However, for NH$_4$Cl-rich mixtures, the band at 1182 cm$^{-1}$ rather than 1186 cm$^{-1}$ was observed at the lowest RH of 4.2%, indicating the formation of SP semisolids due to the uptake of trace amounts of moisture. This suggests that the presence of NH$_4$Cl hindered the crystallization of SP.

The integrated absorbance spanning from 1220 cm$^{-1}$ to 1120 cm$^{-1}$ demonstrates the evolution of SP content during dehumidification in Fig. 5(b). The data indicate values of approximately 0.56, 0.06, and 0.22 for 2:1, 1:1, and 1:2 SP/NH$_4$Cl mixtures, respectively, when the RH reached its minimum value. This implies that the degree of reaction was maximum when SP was mixed with NH$_4$Cl in equal moles, lending credence to the reaction mechanism proposed by Wang et al. in 2019.

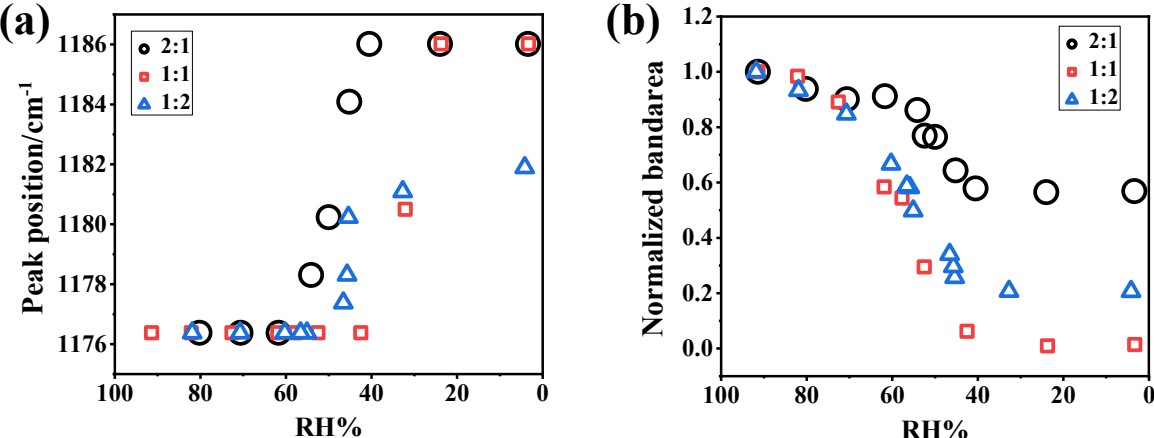

**Figure 5: (a) The peak position shift from 1176 to 1186 cm$^{-1}$ and (b) the integrated areas corresponding integrated band-area for 2:1, 1:1 and 1:2 SP/NH$_4$Cl particles during the dehumidification.**

**3.6 The impact of replacement reaction on organic/ammonium sulfate aerosol phase state**

The component evolution and aerosol phase transition are dependent upon multi processes containing intermolecular interaction, ion mobility, gas-particle partitioning, ion-pair formation, replacement reaction and dissociation, which originated from the composition of the internally mixed aerosols (Li et al., 2021; Mikhailov et al., 2004). In this section, various organic salts, mixed with (NH$_4$)$_2$SO$_4$ were measured to explore the phase behaviors of atmospheric sulfate aerosols. Unlike SP, the applied sodium citrate (SC) and sodium tartrate (ST) exhibit viscous state at lower RH without DRH and ERH.

Fig. 6 displays the IR spectra of SC/(NH$_4$)$_2$SO$_4$ and ST/(NH$_4$)$_2$SO$_4$ particles on dehydration. In Fig. 6(a), as the RH decreases, the bands at 1575 cm$^{-1}$ and 1391 cm$^{-1}$ decreased whereas the 1715 cm$^{-1}$ peak gradually increased, which indicates decrease of the -COO$^-$ and the increase of -COOH. Other observations include the degenerated

band at 1095 cm⁻¹. When the RH decreases to 56.9%, the band at 1132 cm⁻¹ appeared, as well as the band at 995 cm⁻¹ for crystallized sulfate, indicating the crystalline solid $Na_2SO_4$ formation. As the RH further decreased, the 995 cm⁻¹ band for crystalline sulfate became stronger, accompanying the weaker 980 cm⁻¹ absorption peak, suggesting a growing crystalline solid $Na_2SO_4$ content in the particles. While for ST/$(NH_4)_2SO_4$ mixture in Fig. 6(b), we observed the increasing band at 1715 cm⁻¹ and the replacement of 1587 cm⁻¹ band by the 1597 cm⁻¹ band, indicating the transformation from sodium tartrate to sodium bitartrate. In addition, no crystalline solid formed in the aerosols since neither 1132 cm⁻¹ nor 996 cm⁻¹ was observed during the whole dehydration process.

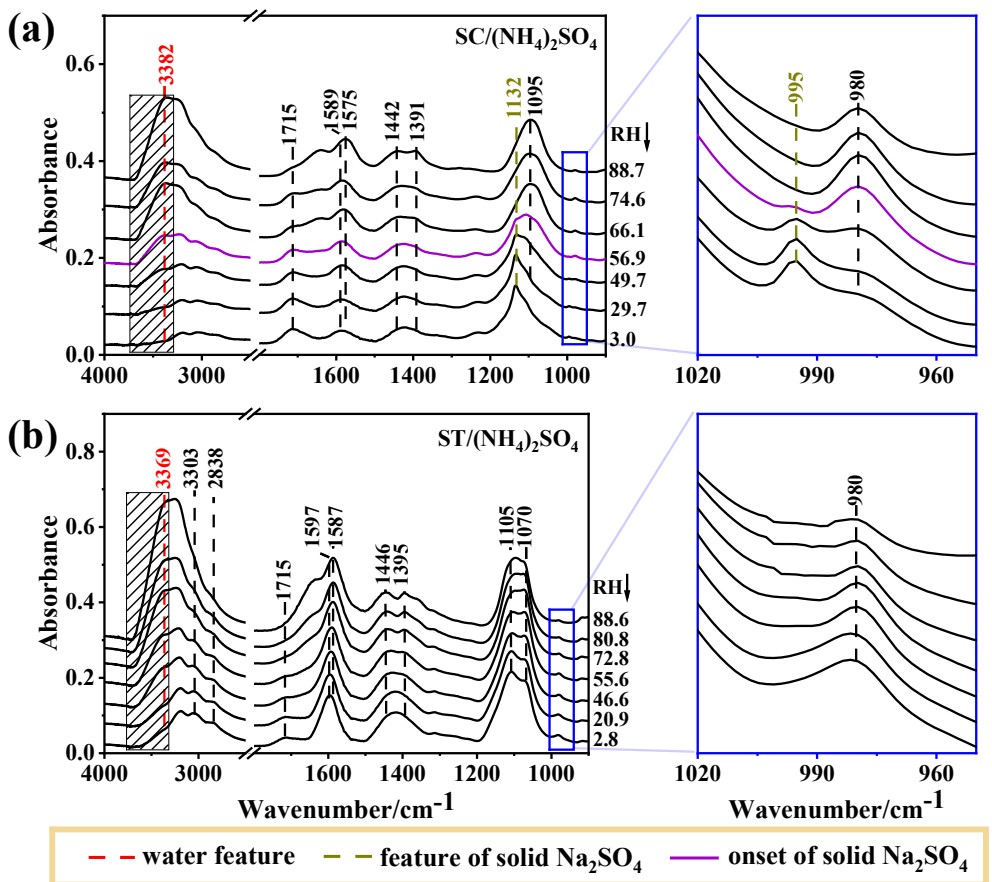

Figure 6: The FTIR spectra of (a) 2:3 SC/(NH₄)₂SO₄ and (b) 1:1 ST/(NH₄)₂SO₄ aerosols on dehydration.

Fig. 7(a) shows the IR spectra as a function of RH for SC/$(NH_4)_2SO_4$ during the hydration process. As the RH increases, the band at 1132 cm⁻¹ maintains a constant intensity below 70.5% RH with a slight enhancement of water features. At 70.5% RH, the band at 1132 cm⁻¹ weakens with a notable increase in the 980 cm⁻¹ band, indicating the onset of $Na_2SO_4$ dissolution. The aerosol, mainly consisting of $Na_2SO_4$, completely deliquesces at 87.2% RH as the band at 996 cm⁻¹ disappears entirely.

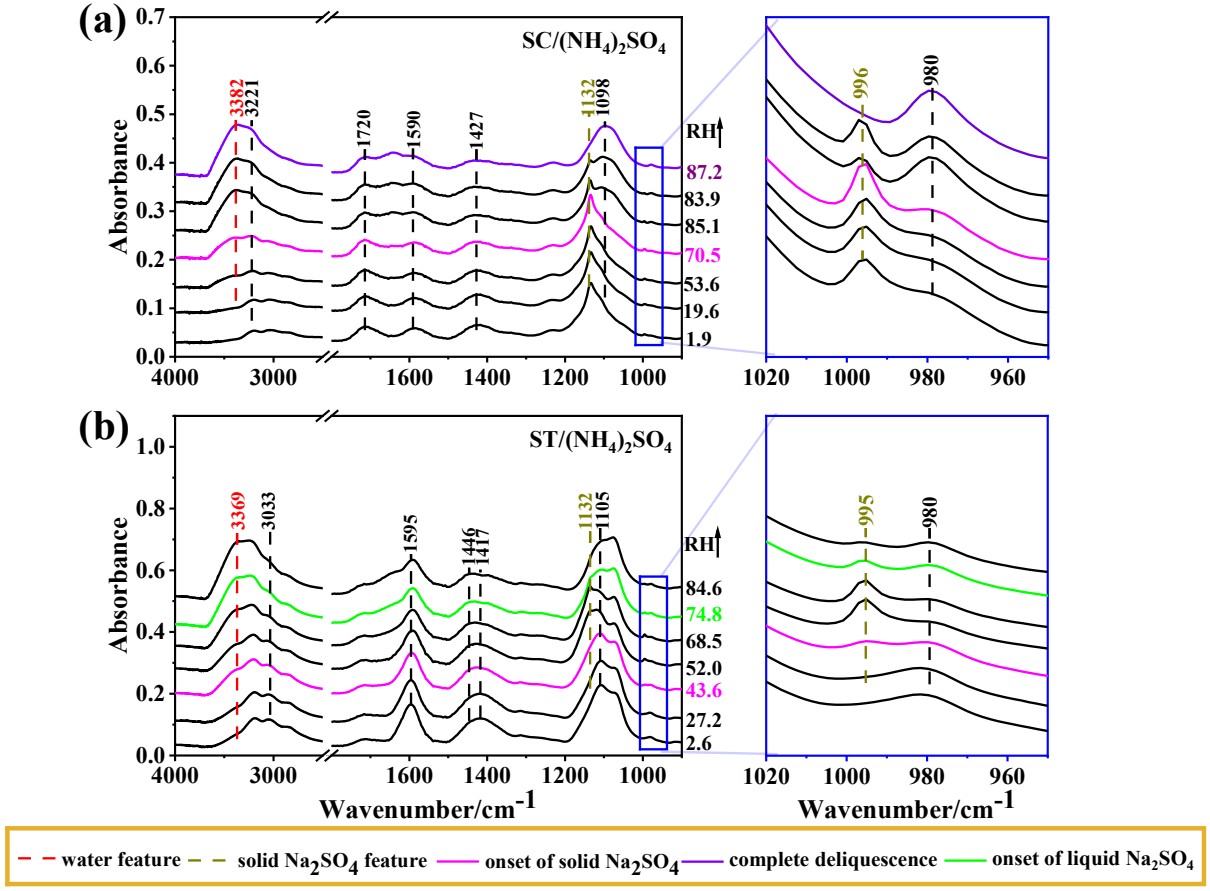

**Figure 7: The FTIR spectra of (a) 2:3 SC/(NH₄)₂SO₄ and (b) 1:1 ST/(NH₄)₂SO₄ aerosols on hydration.**

The IR spectra of ST/(NH₄)₂SO₄ particles are depicted in Fig. 7(b). Upon hydration, bands at 1132 cm⁻¹ and 995 cm⁻¹ are observed at 43.6% RH, suggesting an unexpected formation of crystalline solid Na₂SO₄. As the RH further increases, the IR features of crystalline solid Na₂SO₄ weaken, almost disappears at 84.6% RH, indicating the deliquescence of Na₂SO₄. Similar unconventional behavior of "crystallization on hydration" was also observed for mixed gluconic acid/(NH₄)₂SO₄ and gluconic acid/NaCl aerosols using optical images, where crystalline solid (NH₄)₂SO₄ and NaCl formed upon hydration (Zhu et al., 2022). Likewise, optical images of ST/(NH₄)₂SO₄ were monitored to verify the phase and morphologies during an RH cycle. As shown in Fig. 8, The left-hand side images shows the s round and smooth particle on dehydration, indicative of a homogeneous aqueous state without crystal formed till 12.1%, which is consistent with the absence of 996 cm⁻¹ band in IR spectra (Fig. 6(b)). During humidification, the particles maintain a uniform state with a round shape below 44% RH. At 44% RH, some apparent rectangular dark entities, indicating crystalline solid formation, can be observed within the round particles. Combining the above IR features and morphologies, the aerosols at 44% RH consist of crystalline solid Na₂SO₄, aqueous (NH₄)₂SO₄, and SC. When the RH increases to 80.3% RH, the round and bright sphere is restored,

indicating aerosol deliquescence.

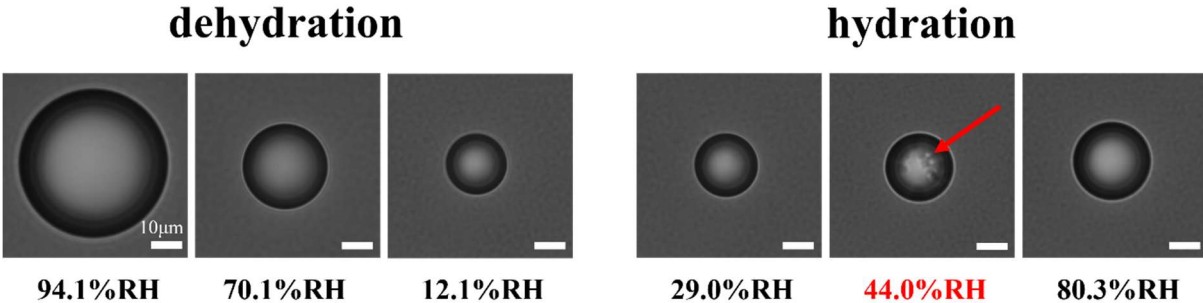

**Figure 8: The optical morphologies of 1:1 ST/(NH₄)₂SO₄ aerosols during a down-up RH cycle. The red arrow pointed the location where the phase transition occurs. The RH indicated in red marks the occurrence of phase transitions. Scale bar represents 10 micrometers.**

As shown in Fig. 6 and Fig. 7, the IR spectra of $SO_4^{2-}$ and organic salts remained distinguishable as the RH changes from ~88.7% to ~2.0% RH stepwise with the rate lower than 5% RH min$^{-1}$, indicating incomplete reaction likely due to mass transfer limitations in viscous aerosol particles for about 30 min at every RH level. With higher viscosity, the degree of chemical reaction tends to decrease. Considering the correlation between the degree of replacement reaction and water content, if the reaction remains incomplete after a RH cycle, the water content in

the solution state during hydration is lower than that during dehydration (Li et al., 2017). The gaps in water content between dehydration and hydration during two RH cycles are illustrated in Fig. S5. For SP/(NH₄)₂SO₄ aerosols, full recovery of water was observed after the second RH cycle, indicative of complete reaction and mainly Na₂SO₄ presence in the particles, consistent with the absence of IR absorptions from (NH₄)₂SO₄ and SP in Fig. S2(a). However, for SC/(NH₄)₂SO₄ and ST/(NH₄)₂SO₄ mixed aerosols, the normal water absorption decreased by 0.1

and 0.14, respectively, after a second RH cycle, providing evidence of the viscous state of the mixed aerosols.

**3.7 The Na₂SO₄ efflorescence upon hydration induced by the replacement reaction**

Pöhlker et al. (2014) applied X-ray microspectroscopy to internally mixed aerosol particles from the Amazonian rainforest collected during anthropogenic pollution and found changes in particle microstructure upon hydration, primarily driven by efflorescence and recrystallization of sulfate salts during aerosol hydration. The efflorescence

upon hydration was attributed to aerosol viscosity and surface tension. Upon hydration, the initially amorphous state of particles is in a metastable state where the formation of the crystalline solid phase is thermodynamically favored but kinetically hindered by nucleation. Continuous water uptake by the particles with rising RH is accompanied by decreasing aerosol viscosity and increasing ion mobility, which could overcome the kinetic

inhibition of ion movement at a certain RH level, leading to nucleation and crystal growth. However, the molecular structure was unknown in Pöhlker's study. Notwithstanding the viscous state for most organics, efflorescence upon hydration is rare. In fact, previous substantial work on the phase state evolution of internally mixed organic/inorganic particles (Wang et al. 2019, Ma et al., 2022, Shao et al., 2018) showed ordinary efflorescence upon dehydration, and to the best of our knowledge, no similar crystallization upon humidification has been reported besides Pöhlker's study. In this work, SC particles retained in a viscous state at lower RH levels as addressed above, while $Na_2SO_4$ crystallized upon dehydration from $SC/(NH_4)_2SO_4$ particles. Therefore, the effect of the molecular structure of organics on inorganic efflorescence needs to be further explored.

According to our recent study on mixture aerosols of gluconic acid and $(NH_4)_2SO_4$ or NaCl, and the $Na_2SO_4$ crystallization from ST and $(NH_4)_2SO_4$ mixtures observed herein, we found that polyhydroxy acids or organic salts containing full-hydroxyl carbon chains is likely causing the efflorescence upon dehydration owing to more viscosity. Grayson et al. (2017) has confirmed viscosity increasing as the number of hydroxyl groups in the molecule. As the RH decreased, the polyhydroxy chain with a carboxyl group is prone to form a gel owing to intermolecular hydrogen bonding. Gels are two-phase mixtures of liquids dispersed in (semi-)solid amorphous matrices, and the uptake of water into a gel can involve gradual swelling as well as stepwise volume increases related to thermodynamically well-defined phase transitions (Pang et al., 2002). When ST was mixed with $(NH_4)_2SO_4$, the polyhydroxy skeleton remained unchanged, and a gel structure gradually formed with the decrease in RH. Based on the fact of the gel state of internally mixed $ST/(NH_4)_2SO_4$ particles, the mechanism for efflorescence upon hydration was proposed (shown in Fig. 9). During the first RH cycle, the ions can transfer freely in the aqueous aerosol. As the RH decreased, tartrate ions and acidified hydrogen tartrate self-assembled into aggregates and gradually gelatinized water due to a decrease in water content, along with $NH_3$ release. Upon drying, gels can form highly porous structures (Li and Gong, 2024; Díaz-Marín et al., 2022). Therefore, in the gel structure, there may exist a collection of fine fibers in these gels where mechanical entanglements create a three-dimensional supramolecular structure to trap water molecules owing to both the chemical adsorption of water molecules onto the gel fibers and physical uptake of water due to capillary condensation (Fig. 9). Due to the strong interaction between OH groups and $SO_4^{2-}$ ions, anions were bound around fibers. Hence, migration of $SO_4^{2-}$ and $Na^+$ ions was inhibited, so that these ions cannot come in contact and nucleate a crystalline phase. The gel structure is a metastable state with a higher energy state than the crystalline state. Upon hydration following dehydration, the initial metastable aerosol can overcome the energy barrier to restore ion mobility in the particle; in turn, the anions and cations can combine to further form a solid nucleus, accompanying continuous $NH_3$ release.

Continuously elevating RH provided more water content in particles, inducing crystal formation from the metastable state surrounded by a more stable solution employed gradually. Further water uptake can overcome the lattice energy to perform a crystalline solid-to-aqueous phase transition, resulting in complete dissolution of crystalline solid $Na_2SO_4$ and aerosol deliquescence.

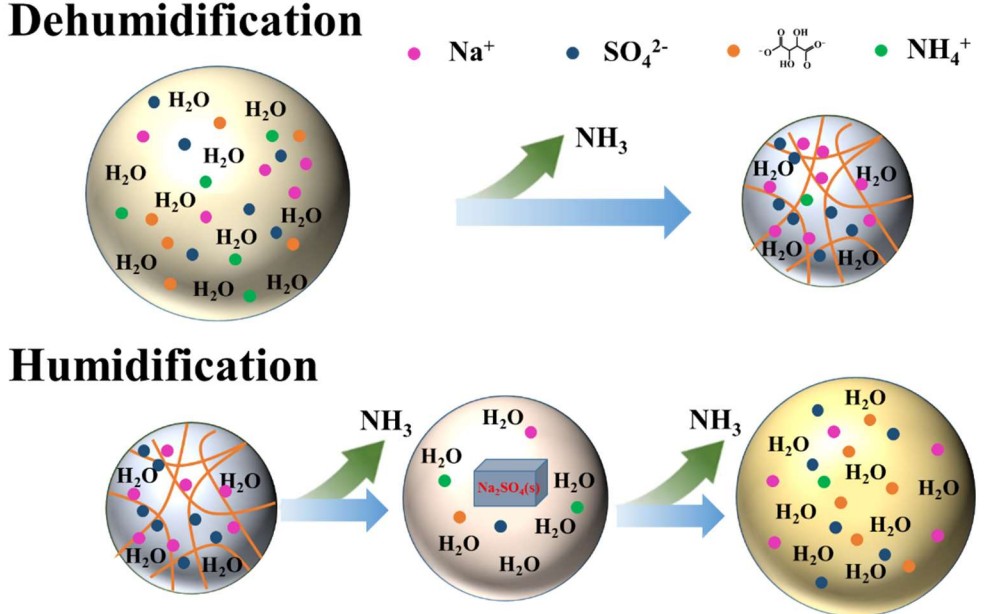

**Figure 9: the schematic phase behavior and chemical process of 1:1 ST/(NH$_4$)$_2$SO$_4$ particles on dehumidification and humidification. The orange curve represents the fibrous gel.**

**4 Conclusions**

Our findings illustrate that the aqueous replacement reaction within internally mixed organic/ammonium aerosols can alter the aerosol composition which in turn affects the phase behavior of aerosol particles. The phase transition behaviors studied in this work are summaried as Table 2. As carboxylic acids and carboxylate salts are abundant in the internal mixtures of atmospheric secondary aerosols (Yang, et al., 2008; Kawamura et al., 2016; Huang et al., 2022), our result implies that the aqueous phase reactions are significant process that can potentially dictate and complicate the phase behavior of internally mixed organic/inorganic aerosols. The change of ERH and DRH, and the occurrence of efflorescence upon humidification, due to the replacement reaction in organic/ammonium aerosols, is crucial for the interaction between aerosols, water vapor, and trace gases, and potentially cause uncertainty for the prediction of atmospheric aerosol phase state. Through the analysis of ATR-FTIR spectra from multiple organic/ammonium aerosol systems under RH cycling, we have provided valuable insights into the complex interactions between organic and inorganic components within aerosol particles.

Table 2 The phase transition behaviors of different organic salts and ammonium salts

| Mixed aerosols | | SP mixed with | | | | | (NH$_4$)$_2$SO$_4$ mixed with | |
|---|---|---|---|---|---|---|---|---|
| | | NH$_4$Cl | | | NH$_4$NO$_3$ | (NH$_4$)$_2$SO$_4$ | SC | ST |
| | | 2:1 | 1:1 | 1:2 | (1:1) | (2:1) | 3:2 | 1:1 |
| ERH% on dehydration | SP | 61.7 | 23.7 | × | NaNO$_3$: 35.7–12.7 | Na$_2$SO$_4$: 65.7–60.1 | Na$_2$SO$_4$: 56.9 | × |
| | NH$_4$Cl | × | 42.5 | 46.6 | SP: × | SP: × | SC: × | |
| DRH% on hydration | SP | 84.3 | × | × | NaNO$_3$: 66.1–82.3 | Na$_2$SO$_4$: 83.8–90.3 | Na$_2$SO$_4$: 70.5–87.2 | Na$_2$SO$_4$: 74.8 |
| | NH$_4$Cl | × | 71.8 | 77.7 | SP: × | SP: × | SC: × | |
| ERH% on hydration | × | × | × | × | × | × | × | Na$_2$SO$_4$: 43.6 |

Note : × means no phase transition was observed

Single value represents the onset ERH or DRH

By examining the hygroscopic behavior and phase transitions of internally mixed particles containing organic acid salts and ammonium salts, we have uncovered the occurrence of replacement reactions and aerosol component depletion during RH changes. Additionally, we observed unconventional crystallization upon hydration behavior for the organic/ammonium mixture aerosols, which can reduce the possibility of further composition evolution and reactive gas uptake due to the absence of ion mobility in solid at higher RH. Hence, light absorbance and CCN activity, which strongly depend on chemical composition, will change. Future studies are needed to further investigate this phenomenon, as it is also expected to occur in other organic/inorganic aerosols that can potentially form gel-like state prior to typical efflorescence.

Our findings highlight the intricate interactions between chemical components of organic/inorganic aerosol (illustrated in Fig. 10). The interplay among organic molecular structure, molar ratio, aqueous replacement reactions, and aerosol phase state can lead to unique and potentially irreversible aerosol evolution during RH cycles. For example, the special chemical process in aerosols induced by gel state formation upon humidification. The observed variations in replacement reactions and product depletion, as well as phase transitions, emphasizes the complexity of aqueous phase aerosol chemistry and its impact on atmospheric processes.

Except for RH, aerosol phase also depends on other atmospheric conditions, particularly temperature. Solid ammonium sulfate particles with organic coatings was observed under high relative humidity (67 to 98%) at low temperature (−2 to +4 °C), underscoring the key role of temperature in phase transitions (Kirpes et al., 2022). They proposed that solid (NH$_4$)$_2$SO$_4$ formed through contact efflorescence and that temperature induced liquid-liquid phase separation (LLPS). Previous studies have shown that temperature often has a greater impact on LLPS

than on efflorescence (Schill et al., 2013). LLPS is crucial for the solidification of ammonium sulfate (Roy et al., 2020) and soot redistribution (Yuan et al., 2023) in aqueous organic-inorganic aerosol droplets. In our work, we observed various aerosol phase states, including crystalline, aqueous, and gel-like states, during dehydration and hydration cycles, and we examined the interplay among aqueous-phase reactions, aerosol hygroscopicity, and phase behavior in detail. However, LLPS was not observed prior to crystallization, either through optical microscopy or ATR-FTIR. Future studies are needed to explore the interplay between various phase transitions, including LLPS, efflorescence, and viscous state formation, considering not only RH but also temperature. By understanding the diverse compositions and phase behaviors of internally mixed organic/inorganic aerosols, we can better address the challenges of air pollution remediation and climate change.

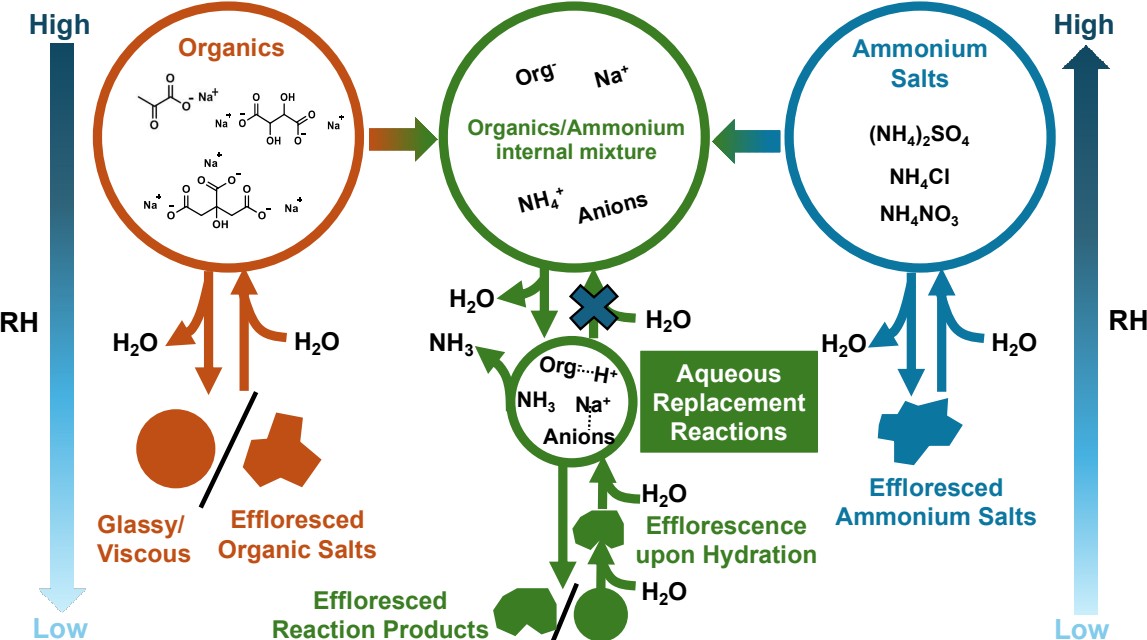

**Figure 10: the schematic of the interplay between aqueous replacement reactions and the phase behavior of internally-mixed organic/inorganic aerosols.**

**Data availability**

Data is available upon request by contacting the corresponding authors.

**Supplement**

The supporting information includes further experimental details and supplementary figures that are associate with the main text.

**Author contributions**

S.P. conceptualized, designed the study. Y.Z., S.P. supervised the study. H.Y., F.D., L.X., conducted the expeirments. H.Y., S.P. analyzed and visualized the data, and worte the orginal draft. Q.H., S.P. reviewed and edited the manuscript.

**Competing interests.**

The authors declare that they have no conflict of interest.

**Acknowledgements**

This study was supported by the Beijing Municipal Natural Science Foundation (No. 8244070), the National Natural Science Foundation of China (No. 91644101), and the Beijing Institute of Technology Research Fund Program for Young Scholars (No. 3100012222337).

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
