# Peer review of "The Interplay between Aqueous Replacement Reaction and the Phase State of Internally Mixed Organic/ammonium Aerosols"

_EGUsphere, 2024_

## Referee Comment (RC1)

Yang and co-workers describe laboratory measurements of internally mixed organic-inorganic aerosol systems. Mainly based on FTIR, their experiments demonstrate how replacement reactions in aqueous aerosols can alter the composition, with subsequent implications for the aerosol phase state, an important property that determines the impacts of aerosol particles on air quality and climate.

Overall, I find the experiments interesting and the topic falls within the scope of Atmospheric Chemistry and Physics. However, I would like the authors to address the points below, before their work can be considered for publication.

L16: Please rewrite and clarify the list of numbers: Do you mean 65.5% to 60.1%?

L17: Please add RH uncertainties to your quoted DRH values. Consider also adding the DRH values of pure inorganics in parenthesis at the end of this sentence, so that the increase in DRH becomes more quantitative.

L29: Please check your reference formatting. This reference is from the year 2011, not 2001.

L33: Please check your reference formatting. This reference should read "Kreidenweis".

L37: "… with carboxylic acids/salts and ammonium salts as one of the most abundant…" Please add appropriate references to this statement.

L44: "Organics can decrease…". Please add appropriate references to this statement, as well as to the following sentence.

L47: Please add more relevant references, e.g.: 10.1021/jp0556759

L50-51: Please either give a review article here as reference, or add further primary refs for each of the named processes, i.e. reactivity, heterogeneous uptake…

L52: This is the first time you mention "replacement reaction". I would encourage the authors to give a brief, general definition and maybe a relevant example, so that a reader can follow more easily.

L54: Check reference: "Ma et al. (Ma et al., 2019)"

L56: "… the impact of … on hygroscopic growth": It would be good to more precisely specify the impact. Did oxalate formation lead to a decrease or increase in hrgoscopic growth?

L57: Check reference: "Ma et al. (Ma et al., 2022)"

L60: Consider changing "liberate" to "release" or "formation"

L64: No need to write Chen et al. twice in the sentence. Rewrite as : "Chen et al. (2020) elucidated…., respectively." Also, is it really appropriate to say "elucidated", since in the next sentence you write how information on aerosol phase state was missing.

L64: I am unclear what "lack of information" or "uncertainty in aerosol phase state" you refer to, please specify? Do you mean a lack of knowing the crystallization RH?

L70-72: This sentence reads cumbersome, please reformulate. I think you are trying to say that phase states such as LLPS can promote the inhomogeneity of components? Your references seems appropriate for the first part ("The inhomogeneity of aerosol components can significantly impact the degree of reactivity"). I would move it there and then consider adding other references for the occurrence of LLPS, liquid-solid coexistence etc.

L76-79: "Therefore, …" I am a bit stuck with this statement. On L69-70 you write that "replacement reactions and phase state are intercorrelated parameters", which I think is a fair way to put it. Now, on this line here you argue about the impact of replacement reactions _on_ aerosol phase state. However, at the same time these replacement reactions to in turn depend on the aerosol phase state. What is the correct line of reasoning, i.e. replacement reactions affecting phase state or phase state affecting replacement reactions…? Maybe it would be fairer to reformulate this here to something like: "Exploring the interplay of different organic and inorganic salts on reactivity, particularly the reactivity of replacement reactions, on the one hand side, and aerosol phase state on the other hand side, is important to understand atmospheric aerosols". The title of the manuscript should be adjusted accordingly.

L77: Change "significance" to "important"

L87: See my comment above: It might be good to replace "correlation" here with "interplay"

L93: What is "three distilled water"?

L97: Please indicate compound purities when introducing the chemicals, along with supplier.

L97: How was this diameter measured? Please specify. Also, were FTIR experiments done for differently sized particles? How was particle homogeneity verified, i.e. absence of e.g. liquid-liquid phase separation, that could affect results.

Table 1: Please add spaces between numeric values and units, e.g. 100 g instead of 100g.

L104: bump → pump

Section 3 general: It would be helpful to add a Table to the SI where the different peaks and the corresponding chemical groups and inferred phase state were summarized, as it is in parts very hard to follow the discussion throughout this Section.

L126: add "is" shifted to

L140: I would encourage to specify here and elsewhere as "crystalline solid phase", as the "viscous phase" can also be solid (== glassy) or semi-solid, which both denote amorphous solid phase states.

L142: Here and elsewhere, please give RH uncertainties.

L145-147: I am unclear about what you write here. If the sodium citrate aerosol is highly viscous, why is the water uptake than not gradual, but "abrupt", as you write? To me the humidification curves on sodium citrate and sodium tartrate look like curves with a sharp deliquescence, which I would only expect for crystalline material. Please clarify why you nonetheless think that these phase states are "viscous", which you seem to use to describe an amorphous solid state.

Section 3: This Section has many different subsections and the connection of these is not always obvious. It would be good to add a brief description of what topic is treated where to guide the reader a bit at the start of the Results and Discussions.

Section 3.1 general: It could make sense to move your Fig. S2 to the main text. That way, a reader could follow the described spectral changes with changes in RH better. Also, Fig. S2 misses a proper legend indicating what the black and red data points correspond to.

L153-155/Fig.1: In the blue-framed enlarged image: Where is the blue and the red lines? Please color your lines as in the left most panel or the violet-framed enlarged image. Otherwise

it is impossible to connect the RH values to the individual lines. Note, this also concerns Fig. 1c.

L159: Reference formatting: No need to have "Tan et al." twice in this sentence, please change.

L185-195: Please add appropriate references the subpanels of Fig. S3, to make it clearer what you are describing here.

Section 3.3 general: The structure of this section should be improved. In the beginning, the authors introduce relevant replacement reactions. The middle part (L202-210 then discusses some literature findings. The last part (L210-235) then discuss hygroscopicity data from the present manuscript, but the link to the topic of replacement reactions remains unclear. This requires improvement in a revised version.

L198-200: Please indicate phase states of reactants too. Please also indicate whether these reactions take place for decreasing or increasing RH.

L206: "larger specific surface area" compared to what? Bulk solutions? Unclear.

L215: "…water loss or uptake". Add https://doi.org/10.1021/cr990034t or other appropriate refs.

L221: "These processes…": Can you quantify "sudden" here? Also, please elaborate how spectral changes in this case do not reflect the water content. What does this mean to the spectral changes in your Fig. 1 (and others) discussed above?

Fig. 2: Please add labels for the red and pink shaded area directly into the Fig. to make interpretation easier.

L224-241:

- Please add references to appropriate subpanels in Fig. 2 directly to the text, as you jump quickly between the individual panels.
- Please add information on the rate of RH changes during your experiments to the text. Was the rate slow enough to allow for equilibration of the particles and the surrounding RH?

L261: Check title, should this read: "The effect of molar ratio on the replacement…"

L265: Please provide reasoning for the chosen ratio in the context of typical organic-to-inorganic ratio found in atmospheric aerosols.

L267-270: I suggest to repeat in parenthesis the meaning of the named peaks at e.g. 3130 cm-1 etc., to make it easier for the reader to follow here.

L278: "From Fig. 4a…" Looking at the red squares in Fig. 4a, the shift from 1176/cm to 1186/cm appears to be pretty broad from about 40% RH to 20% RH. How can you get such a precise ERH value as 41% for the 1:1 mixture? Also, for the 2:1 mixture the peak shift seems to appear over an even broader RH range, so how do you get to the quoted 23.7% RH? Please clarify in the text.

Fig. 4: What is the difference between grey symbols with a horizontal vs. vertical grey line across the symbol?

L294: Please delete "et al."

L295: mixing → mixed, was → were

L303: Replace "escalated" with more appropriate wording.

Fig. 7: Why was there no crystallization observed in the left-hand side images at 12.1% RH? Please clarify in text.

L332-341: More information should be added how these experiments were done? What was the rate of RH changes? Was the first RH cycle started at high or low RH? How long were the particles exposed to these (high) RH conditions? The latter would help to get a better idea how fast these replacement reactions are.

L352: "substantial work" should be followed by appropriate refs.

L354: "retained in a viscous"

L359: Please check if you really mean "dehydration" here or "hydration"?

L360: "Grayson…" verb is missing. Please avoid repetition of references; otherwise check for correct punctuation, i.e. "Grayson et al. have demonstrated..."

L361: "As the RH decreased…" Please add ref to this statement and provide a brief explanation of "gel" for the reader here (you could move the explanation you give on L367 up).

L371: which → where

L372: How does this trapping work? Just by physical uptake of water due to capillary condensation or is this due to chemisorption? This should be added to the text.

L373: Break up sentence: "… bound around fibres. Hence, migration of SO42- and Na+ ions was inhibited, so that these ions cannot come in contact and nucleate a crystalline phase."

Fig. 8: I like the idea of having a schematic, but I think this figure can be improved to clarify aspects described in the text. It is unclear what the different colors correspond to. A legend is missing to indicate the fibrous particles as "gel-like". Is there a better way to make the "OH groups and SO42- ions bound around the fibers" (L372) more obvious? Outgassing of NH3 is depicted but not mentioned in the text.

L387: "of atmospheric SOA." Please add more appropriate refs.

L401-404: Please see my previous comment: Replacement reactions can certainly impact the aerosol phase state as you document, but in turn these are also dependent on the phase state of the aerosol particle. This aspect should be better represented in your Conclusion section.

L397-398: "Additionally, we observed…" This is certainly a very interesting finding of this study. I would like to see some more discussion, which atmospheric processes could be influenced by such "crystallization upon hydration". Also, can the authors speculate how important this process is and if they would expect it for other atmospherically relevant aerosol systems?

L405-409: I am unclear what "targeted strategies to mitigate air pollution" the authors refer to. Please elaborate or remove this statement.

Fig. 9: This figure is not mentioned and discussed in the text, please do that or delete the figure.

---

## Author Comment (AC1)

**Response to referee 1's comments**

In Yang and co-workers describe laboratory measurements of internally mixed organic-inorganic aerosol systems. Mainly based on FTIR, their experiments demonstrate how replacement reactions in aqueous aerosols can alter the composition, with subsequent implications for the aerosol phase state, an important property that determines the impacts of aerosol particles on air quality and climate.

Overall, I find the experiments interesting and the topic falls within the scope of Atmospheric Chemistry and Physics. However, I would like the authors to address the points below, before their work can be considered for publication.

**Author reply:** We are grateful for the careful review, the positive feedback, and the constructive suggestions from the referee. These suggestions are of great significance in enhancing the quality of our manuscript and facilitating readers' better understanding of our work. We have incorporated the referee's comments and suggestions into our manuscript. In the response to the referee, we list each comment, followed by our detailed responses, corresponding revisions with accompanying line numbers, and additional figures. We believe that the changes have resulted in a much stronger manuscript.

**1L16: Please rewrite and clarify the list of numbers: Do you mean 65.5% to 60.1%?**

**Reply:** Thanks for the careful review. We have carefully checked the manuscript and corrected this error.

**Revision: line 15-17:** "For SP/ammonium aerosols, the produced $NaNO_3$ and $Na_2SO_4$ crystallized from 35.7% to 12.7%, and from 65.7% to 60.1% RH, respectively, lower than pure inorganics ($62.5\pm9-32\%$ RH for $NaNO_3$ and $82\pm7-68\pm5\%$ RH for $Na_2SO_4$)."

**L17: Please add RH uncertainties to your quoted DRH values. Consider also adding the DRH values of pure inorganics in parenthesis at the end of this sentence, so that the increase in DRH becomes more quantitative.**

**Reply:** Thank you for this helpful suggestion. We have added the uncertainties and the DRH of pure $Na_2SO_4$ and $NaNO_3$.

**Revision: line 17-19:** "Upon hydration, the crystalline $Na_2SO_4$ and $NaNO_3$ deliquesced at 88.8%–95.2% and $76.5\pm2-81.9\%$, higher than those of pure $Na_2SO_4$ ($74\pm4\%-98\%$ RH) and $NaNO_3$ ($65-77.1\pm3\%$ RH)."

**L29: Please check your reference formatting. This reference is from the year 2011, not 2001.**

**Reply:** Thank you for pointing this out. This has been corrected in the revised manuscript.

**Revision: line 32-33:** "…water (Koop et al., 2011),"

**L33: Please check your reference formatting. This reference should read "Kreidenweis".**

**Reply:** Thank you for pointing this out. We have corrected the reference format and checked the format for other for our manuscript.

**Revision: line 34:** "…(Svenningsson et al., 1997; Kreidenweis et al., 2005),"

**L37: "… with carboxylic acids/salts and ammonium salts as one of the most abundant…" Please add appropriate references to this statement.**

**Reply:** Two references have been added according to the referee's suggestion.

**Revision: line 39-40:** "…components, respectively (Huang et al., 2014; Trebs, et al., 2005)."

line 501-505: "Huang, R.-J., Zhang, Y., Bozzetti1, C., Ho, K.-F., Cao, J.-J., Han, Y., Daellenbach, K. R., Slowik, J. G., Platt, S. M., Canonaco, F., Zotter, P., Wolf, R., Pieber, S. M., Bruns, E. A., Crippa, M., Ciarelli, G., Piazzalunga, A., Schwikowski, M., Abbaszade, G., Schnelle-Kreis, J., Zimmermann, R., An, Z., Szidat, S., Baltensperger, U., Haddad, I. E., Prévôt, A. S. H.: High secondary aerosol contribution to particulate pollution during haze events in China, Nature, 514, 218-222, http://doi:10.1038/nature13774, 2014."

line 617-620: Trebs, I.; Metzger, S.; Meixner, F. X.; Helas, G. N.; Hoffer, A.; Rudich, Y.; Falkovich, A. H.; Moura, M. A. L.; da Silva, R. S.; Artaxo, P.; Slanina, J.; Andreae, M. O. The $NH_4^+$-$NO_3^-$-$Cl^-$-$SO_4^{2-}$-$H_2O$ aerosol system and its gas phase precursors at a pasture site in the amazon basin: How relevant are mineral cations and soluble organic acids? J. Geophys. Res.-Atmos. 110 (D7), D07303, http://doi.org/10.1029/2004JD005478, 2005.

**L44: "Organics can decrease…". Please add appropriate references to this statement, as well as to the following sentence.**

**Reply:** Thank you very much for your suggestion. Two references have been added.

**Revision: line 47-49:** "Organics can decrease the surface activity of particles due to hydrophobic carbon chains and hydrophilic head groups (Petters et al., 2007). Additionally, viscous states owing to intermolecular interactions at low humidity can be formed, leading to limited water absorption (Reid et al., 2018 )."

line 578-580: "Petters. M. D., Kreidenweis, S. M.: A single parameter representation of hygroscopic growth and cloud condensation nucleus activity, Atmos. Chem. Phys. 7,1961-1971, https://doi.org/10.5194/acp-7-1961-2007, 2007."

line 591-593: "Reid, J. P., Bertram, A. K., Topping, D. O., Laskin, A., Martin, S. T., Petters, M. D., Pope, F. D., Rovelli, G.: The viscosity of atmospherically relevant organic Particles, 9, 956, https://doi.org /10.1038/s41467-018-03027-z, 2018"

**L47: Please add more relevant references, e.g.: 10.1021/jp0556759**

**Reply:** Thank you. The reference has been added in revised paper.

**Revision: line 49-50:** "…organic acids and inorganic salts (Shi et al., 2012; Wang et

al., 2017; Marcolli et al., 2006)"

line 554-556: "Marcolli, C., Kreidenweis, U. K.: Phase Changes during Hygroscopic Cycles of Mixed Organic/Inorganic Model Systems of Tropospheric Aerosols, J. Phys. Chem. A, 110, 1881–1893, https://doi.org/10.1021/jp0556759, 2006."

**L50-51: Please either give a review article here as reference, or add further primary refs for each of the named processes, i.e. reactivity, heterogeneous uptake…**

**Reply:** Thank you for your suggestion, we cited a review article as our reference.

**Revision: line 53:** "…trace gases (Herrmann et al., 2015)."

**Line 495-497:** "Herrmann, H., Schaefer, T., Tilgner, A., Styler, S. A., Weller, C., Teich, M.; Otto, T.: Tropospheric Aqueous-Phase Chemistry: Kinetics, Mechanisms, and Its Coupling to a Changing Gas Phase, Chem. Rev. 115, 4259-4334, https://doi.org/10.1021/cr500447k, 2015."

**L52: This is the first time you mention "replacement reaction". I would encourage the authors to give a brief, general definition and maybe a relevant example, so that a reader can follow more easily.**

**Reply:** We thank the referee for the suggestion. The definition and an example of replacement reaction has been added in revised paper and highlighted.

**Revision: line 56-58:** "Here and throughout this paper, we refer the replacement reaction to the exchange of ions between two ionic compounds (e.g., $CH_3CH_2Br$ + $NaOH \rightarrow NaBr + CH_3CH_2OH$)."

**L54: Check reference: "Ma et al. (Ma et al., 2019)"**

**Reply:** Thanks, it has been corrected.

Revision: line 58: "Ma et al. (2019)…"

**L56: "… the impact of … on hygroscopic growth": It would be good to more precisely specify the impact. Did oxalate formation lead to a decrease or increase in hygroscopic growth?**

**Reply:** Thank you for the comment. This sentence has been revised to improve clarity.

**Revision: line 59-60:** "…validated that the metal oxalate complex formation led to a decrease in hygroscopic growth."

**L57: Check reference: "Ma et al. (Ma et al., 2022)"**

**Reply:** I have checked the ref. and corrected the reference.

Revision: line 61: "Ma et al. (2022)…"

**L60: Consider changing "liberate" to "release" or "formation"**

**Reply:** Thank you. We have revised the sentence and removed this expression.

**L64: No need to write Chen et al. twice in the sentence. Rewrite as : "Chen et al. (2020) elucidated…., respectively." Also, is it really appropriate to say "elucidated", since in the next sentence you write how information on aerosol phase state was missing.**

**Reply:** Thanks, the reference has been corrected. In addition, we replace "elucidated" with "discussed" in this sentence for a more objective expression.

Revision: line 66: "…Chen et al. (2022) discussed …".

**L64: I am unclear what "lack of information" or "uncertainty in aerosol phase state" you refer to, please specify? Do you mean a lack of knowing the crystallization RH?**

**Reply:** Thank you for your comment. The sentence has been revised for clarification. In the cited article, due to the lack of crystallization RHs of the produced salts, the thermodynamics and kinetic analysis on depletion of chloride, nitrate, or ammonium in atmospheric aerosols were discussed without considering crystallization RHs of the produced salts. In the original sentence, we refer the "lack of information" to the lack of crystallization of RH of reaction products, and the "uncertainty" was referred to limited knowledge about possible gel or amorphous phase formation, which was also not discussed in the cited paper.

Revision: line 68-69: "However, due to the lack of crystallization RH of each replacement reaction products, and the limited knowledge about potential gel or amorphous phase formation, the kinetic processes based on the model cannot accurately take aerosol phase state into account."

**L70-72: This sentence reads cumbersome, please reformulate. I think you are trying to say that phase states such as LLPS can promote the inhomogeneity of components? Your references seems appropriate for the first part ("The inhomogeneity of aerosol components can significantly impact the degree of reactivity"). I would move it there and then consider adding other references for the occurrence of LLPS, liquid-solid coexistence etc.**

**Reply:** Thank you for the suggestions. The sentence has been revised for clarity and a reference was also added.

**Revision: line 71-74:** "Reactions in atmospheric aerosols, such as replacement reaction, can alter aerosol phase state. Atmospheric aerosol phase states can be complex, including the coexistence of crystalline solid and liquid (Yang et al., 2019), liquid-liquid phase separation (Zhou et al., 2019), and solid in a viscous phase. (Zhu et al., 2022). These varying phase states promote the inhomogeneity of aerosol components, significantly impacting the degree of reactivity. (Zong et al., 2022)."

**Line 651-653:** "Zhou, S., Hwang, B. C. H., Lakey, P. S. J., Zuend, A., Abbatt, J. P. D., Shiraiwa, M.: Multiphase reactivity of polycyclic aromatic hydrocarbons is driven by phase separation and diffusion limitations, 24, 11658-11633, https://doi.org/10.1073/pnas.1902517116, 2019."

**L76-79: "Therefore, …" I am a bit stuck with this statement. On L69-70 you write that "replacement reactions and phase state are intercorrelated parameters", which I think is a fair way to put it. Now, on this line here you argue about the impact of replacement reactions *on* aerosol phase state. However, at the same time these replacement reactions to in turn depend on the aerosol phase state. What is the correct line of reasoning, i.e. replacement reactions affecting phase state or phase state affecting replacement reactions…? Maybe it would be fairer to reformulate this here to something like: "Exploring the interplay of different organic and inorganic salts on reactivity, particularly the reactivity of replacement reactions, on the one hand side, and aerosol phase state on the other hand side, is important to understand atmospheric aerosols". The title of the manuscript should be adjusted accordingly.**

**Reply:** Thank you very much. The sentence has been rewritten as your recommendation. And the title of the manuscript has been adjusted.

**Revision: line 77-79:** "… exploring the interplay of different organic and inorganic salts on reactivity, particularly the reactivity of replacement reactions, and aerosol phase state on the other hand, is important to understand atmospheric aerosols."

**Line 1-3:** "The Interplay between Aqueous Phase Replacement Reaction and the Phase State of Internally Mixed Organic/ammonium Aerosols"

**L77: Change "significance" to "important"**

**Reply:** Thank you. The word "significance" has been changed to "important" which can be seen in last reply.

**Line 77-79:** "is important to understand atmospheric aerosols"

**L87: See my comment above: It might be good to replace "correlation" here with "interplay"**

**Reply:** Thanks, we have replaced correlation with "interplay".

**Revision: line 89:** "…a better understanding of the interplay between chemical composition…"

**L93: What is "three distilled water"?**

**Reply:** Thank you. The "three distilled water" means that the water was distilled three times. It has been corrected as "triple distilled water"

**Revision: line 95-96:** "…were prepared using triple distilled water."

**L97: Please indicate compound purities when introducing the chemicals, along with supplier.**

**Reply:** Thanks for your suggestion. The purity and supplier has been added in revised paper.

**Revision: line 96-99:** "The organic acid salts include sodium pyruvate (SP, Aladdin

Reagent Co., Ltd., ≥99.0%), sodium citrate (SC, Beijing Chemical Reagents Company, ≥99.0%), and sodium tartrate (ST, Beijing Chemical Reagents Company, ≥99.0%). The ammonium salts include $(NH_4)_2SO_4$, $NH_4NO_3$ and $NH_4Cl$ (all from Beijing Chemical Reagents Company, ≥99.0%)."

**L97: How was this diameter measured? Please specify. Also, were FTIR experiments done for differently sized particles? How was particle homogeneity verified, i.e. absence of e.g. liquid-liquid phase separation, that could affect results.**

**Reply:** We thank the reviewer for the comment. The dimeter was measured by optical microscopy. In our experiment, droplets were deposited onto the substrate and kept at the highest level (>87% RH) for about 30 min before size measurement. The droplet size distribution follows a Gaussian distribution with a mean of 5 micrometers. The homogeneity of the droplets was also verified according to optical images. No LLPS was observed at relatively high RH prior to efflorescence.

**Table 1: Please add spaces between numeric values and units, e.g. 100 g instead of 100g.**

**Reply:** Thanks, we have checked the whole paper and add spaces between numeric values and units. The revised parts have been highlighted.

**Revision:**

**Table 1: The molecular structure of inorganic and organic compounds**

| Compound | Molecular structure | Dissociation constant of conjugate acid | Solubility in 100g H$_2$O (20°C) | Molecular weight (Da) |
|---|---|---|---|---|
| Ammonium nitrate | $NH_4NO_3$ | **NA** | 190 g | 80.043 |
| Ammonium chloride | $NH_4Cl$ | **NA** | 37.2 g | 53.49 |
| Ammonium sulfate | $(NH_4)_2SO_4$ | **NA** | 75.4 g | 132.14 |
| Sodium pyruvate |  | $K_a = 3.2 \times 10^{-3}$ | 47 g | 82.03 |

| Compound | Structure | $K_a$ values | Solubility | MW |
|---|---|---|---|---|
| Sodium citrate | | $K_{a,1} = 7.4 \times 10^{-4}$
$K_{a,2} = 1.7 \times 10^{-5}$
$K_{a,3} = 4.0 \times 10^{-7}$ | 154 g | 258.07 |
| Sodium tartrate | | $K_{a,1} = 1.04 \times 10^{-3}$
$K_{a,2} = 4.55 \times 10^{-5}$ | 33.3 g | 194.05 |

**L104: bump → pump**

**Reply:** The "bump" has been replaced by "pump".

**Revision: line 107:** "…mass flowmeter and vacuum pump…"

**Section 3 general: It would be helpful to add a Table to the SI where the different peaks and the corresponding chemical groups and inferred phase state were summarized, as it is in parts very hard to follow the discussion throughout this Section.**

**Reply:** We thank the reviewer for the helpful suggestion. Table S1 has been added, as well as the statement in paper.

**Revision: line 145-146:** "…and band assignments can be found in Table S1"

**Table S1** Assignments of IR Peaks Attributed to Ammonium and Carboxylates

| | | | | | | Assignment |
|---|---|---|---|---|---|---|
| | | | 831 | | | $\nu_2$-$NO_3^-$ |
| | | 1055(s) | | | | $\gamma(CH_2)$ |
| | | 1069(l) | | | | $\gamma(CH_2)$ |
| 1176(l) | | | | | | $\nu$ (C-C) + $\gamma$ (COO⁻)(liquid) |
| | | | | | 1180 | |
| 1186(s) | | | | | | $\nu_3(SO_4^{2-})$ |
| | 1278 | 1276(s) | | | | $\nu$ (C-C) + $\gamma$ (COO⁻) |
| | 1308 | | | | | $\nu$(C-C) |
| 1353 | | | | | | $\omega(CH_2)$ |
| | 1386 | 1388(l) | | | | $\delta_s$ (CH$_3$) |
| 1404(s) | | | 1417(s) | 402(s) | 1412(s) | $\nu_s$ (COO⁻) |
| | | | | | | $\nu_2(NH_4^+)$(solid) |
| 1424(l) | | 1436(s) | 1448(l) | 1440(l) | 1443(l) | $\nu_s$ (COO⁻) |

| | | |
|---|---|---|
| 1568 | | $\nu_2(NH_4^+)$(liquid) |
| 1608(l) | | $\nu_{as}$ (COO$^-$) |
| 1627(s) | | $\delta$ (OH) + $\nu$ (CO) |
| 1654(s) | | $\nu_s$ (COO$^-$) |
| 1709 | | $\nu_s$ (COO$^-$) |
| | | $\nu_{as}$ (COO$^-$) |
| | 1754 | $\nu_{as}$ (C=O) |
| | 2804 | |
| | 3036 | $\nu$ (NH$_4^+$) |
| 3360 | 3130 | $\nu$ (NH$_4^+$) |
| | | $\nu$ (NH$_4^+$) |
| | | $\nu_{as}$(O–H) |

s: solid state, l: liquid state

**L126: add "is" shifted to**

Reply: Thank for your careful reading, the "is" has been added.

Revision: line 149: "1176 cm$^{-1}$ is shifted to 1186 cm$^{-1}$"

**L140: I would encourage to specify here and elsewhere as "crystalline solid phase", as the "viscous phase" can also be solid (== glassy) or semi-solid, which both denote amorphous solid phase states.**

Reply: The advice is very good. We have specified the "crystalline solid phase" in appropriate place to avoid the confusion with amorphous solid phase states, all of which have been highlighted.

**L142: Here and elsewhere, please give RH uncertainties.**

Reply: In revised paper, the RH uncertainties have been added in some places and highlighted.

Revision: line 168: "…at 55.7−59% RH for sodium tartrate aerosols, where an error margin is 1%."

**L145-147: I am unclear about what you write here. If the sodium citrate aerosol is highly viscous, why is the water uptake than not gradual, but "abrupt", as you write? To me the humidification curves on sodium citrate and sodium tartrate look like curves with a sharp deliquescence, which I would only expect for crystalline material. Please clarify why you nonetheless think that these phase states are "viscous", which you seem to use to describe an amorphous solid state.**

**Reply:** The viscous state was concluded by little water retained at the lowest RH, which can be seen from the water content dependent upon the RH during dehydration as seen in Fig. 1. The more detailed description has been added. On hydration, the viscous state can transform into solution at a certain RH, which is gel-broken point.

**Revision: line 163-164:** "…and the retained normal water content of 0.25 and 0.15 for ST and SC, respectively, at the lowest RH…"

**Section 3: This Section has many different subsections and the connection of these is not always obvious. It would be good to add a brief description of what topic is treated where to guide the reader a bit at the start of the Results and Discussions.**

**Reply:** Based on referee's suggestion. A brief description that guides the reader to different topics was added at the start of the Results and Discussions.

**Revision: line 123-141:** "The phase behaviour of mixed aerosols is dependent upon the chemical composition, molar ratio and chemical process. Herein we selected three organic salts including SP, ST and SC, along with $NH_4Cl$, $NH_4NO_3$ and $(NH_4)_2SO_4$, to study the interplay between phase state, composition evolution, and aqueous phase replacement reactions in aerosols during RH changing cycles. This section consists of seven parts. In *part 3.1*, we analysed the IR spectra of pure organic salts and inorganic salts to obtain the differences in characteristic peaks for aqueous and solid phases. Based on this, in *part 3.2*, SP was mixed with different ammonium salts with various stoichiometric ratio to form aerosols, and the infrared spectra of the mixed aerosols were tested to analyse the characteristic absorption peaks in the second section. The results showed that SP underwent a substitution reaction with the ammonium salts, and the mixed aerosols containing different ammonium salts exhibited distinct phase behaviours. Therefore, in *parts 3.3 and 3.4*, we discussed in detail the substitution reactions and the resulting changes in water content and phase transition regions of the compounds. The phase transition regions of the reaction products were compared with those of the pure components. The relative content of the mixed components may vary in different regions. In *part 3.5*, we took the SP/NH4Cl system as an example to study the impact of different component contents on phase transition behaviour. After investigating the hygroscopic behaviour of mixed aerosols containing different ammonium salts with the same organic salt, in *part 3.6*, we examined the phase evolution of aerosols induced by mixing different organic salts, SC and ST, with ammonium sulphate. By combining infrared spectroscopy and optical images, we discovered that the mixture of tartaric acid and ammonium sulphate not only underwent substitution reactions but also exhibited unexpected hygroscopic weathering behaviour. Therefore, in *part 3.7*, we discussed the causes of this special phase behaviour."

**Section 3.1 general: It could make sense to move your Fig. S2 to the main text. That way, a reader could follow the described spectral changes with changes in RH better. Also, Fig. S2 misses a proper legend indicating what the black and red data points correspond to.**

**Reply:** Thanks for the advice. Fig. S2 have been moved to the main text as Fig. 1 and

a proper legend has been added.

[Figure]

Fig. 1. The and IR spectra of organic salts on dehydration and hygroscopic behavior during a down-up RH cycle. The shaded area shows the chosen integration region for liquid water. The spectra for sodium pyruvate was previous reported by Yang et al (2019).

**L153-155/Fig.1: In the blue-framed enlarged image: Where is the blue and the red lines? Please color your lines as in the left most panel or the violet-framed enlarged image. Otherwise, it is impossible to connect the RH values to the individual lines. Note, this also concerns Fig. 1c.**

**Reply:** Thanks for the advice. The lines in the left panel and right enlarged curves are correspondingly matched in the revised Fig. 1.

**Revision:**

[Figure]

**L159: Reference formatting: No need to have "Tan et al." twice in this sentence, please change.**

**Reply:** Thank you. This has been corrected in revised paper.

**Revision: line 188:** "…RH. Tan et al (2014) investigated…"

**L185-195: Please add appropriate references the subpanels of Fig. S3, to make it clearer what you are describing here.**

**Reply:** Thank you. In revised version, a sentence to describe Fig. S2 (Fig. S3 in the original SI) has been added.

**Revision: line 211-212:** "Following the dehydration, the hydration of three SP/ammonium aerosols was performed. Fig. S2 presents the IR spectra on hydration."

**Section 3.3 general: The structure of this section should be improved. In the beginning, the authors introduce relevant replacement reactions. The middle part (L202-210 then discusses some literature findings. The last part (L210-235) then discusses hygroscopicity data from the present manuscript, but the link to the topic of replacement reactions remains unclear. This requires improvement in a revised version.**

**Reply:** Thank you for your suggestion. We believe that the whole paragraph has been improved following your suggestion. We have revised Section 3.3 including the topic and the first paragraph.

**Revision: line 228-242:** " **3.3 The replacement reactions in SP/ammonium aerosols and the resulting hygroscopicity**

Above analysis on IR features showed the following reactions (1) - (3) in mixed SP/ammonium aerosols:

$$CH_3COCOONa(aq) + (NH_4)_2SO_4(aq) \xrightarrow{RH\ decreasing} CH_3COCOOH(g) + NH_3(g) + Na_2SO_4(s) \qquad (1)$$

$$CH_3COCOONa(aq) + NH_4Cl(aq) \xrightarrow{RH\ decreasing} CH_3COCOOH(g) + NH_3(g) + NaCl(s) \qquad (2)$$

$$CH_3COCOONa(aq) + NH_4NO_3(aq) \xrightarrow{RH\ decreasing} CH_3COCOOH(g) + NH_3(g) + NaNO_3(s) \qquad (3)$$

It can be seen that pyruvic acid and ammonia are formed and depleted from particles alongside the crystalline solid formation of various inorganic salts during dehydration, which proceeded the replacement reaction between SP and ammonium. Herein, aerosol particles act as micro-reactors, with their larger specific surface area than bulk solution facilitating similar processes. In fact, the replacement reactions in aerosols driven by gas released or compound formation with lower solubility have been reported in previous work. For example, Wang et al. (2017) observed the reaction in aerosols composed of oxalic acid and ammonium sulfate, which was driven by formation of lower hygroscopic ammonium hydrogen oxalate ($NH_4HC_2O_4$) and ammonium hydrogen sulfate ($NH_4HSO_4$) during the dehydration process. While Wang et al. (2019) reported the fact of the formation of crystalline solid $Na_2SO_4$ from $(CH_2)_n(COONa)_2$ (n = 1, 2)/$(NH_4)_2SO_4$ aerosols upon dehydration. Building upon their findings, crystalline solid $NaNO_3$ and $NaCl$ are also formed, as shown in equations (2) and (3).

**L198-200: Please indicate phase states of reactants too. Please also indicate whether these reactions take place for decreasing or increasing RH.**

**Reply:** Thank you for the suggestion. The phase states of reactants and reaction condition have been added in revised paper.

**Revision: line 230-232:**

$$CH_3COCOONa(aq) + (NH_4)_2SO_4(aq) \xrightarrow{RH\ decreasing} CH_3COCOOH(g) + NH_3(g) + Na_2SO_4(s) \qquad (1)$$

$$CH_3COCOONa(aq) + NH_4Cl(aq) \xrightarrow{RH\ decreasing} CH_3COCOOH(g) + NH_3(g) + NaCl(s) \qquad (2)$$

$$CH_3COCOONa(aq) + NH_4NO_3(aq) \xrightarrow{RH\ decreasing} CH_3COCOOH(g) + NH_3(g) + NaNO_3(s) \qquad (3)$$

**L206: "larger specific surface area" compared to what? Bulk solutions? Unclear.**

**Reply:** The aerosol particles are microdroplets, which has larger specific surface area than bulk solution under the condition of the same volume. In revised paper, the sentence has been corrected.

**Revision: line 235:** "…with their larger specific surface area than bulk solution facilitating…"

**L215: "…water loss or uptake". Add https://doi.org/10.1021/cr990034t or other appropriate refs.**

**Reply:** Thanks. the literature has been added in revised paper and highlighted.

**Revision: line 247:** "…water loss or uptake (Martin, 2000)"

**Line 557-558** "Martin, S. T.: Phase Transitions of Aqueous Atmospheric Particles, Chem. Rev., 100, 3403–3453, https://doi.org/10.1021/cr990034t, 2000."

**L221: "These processes…": Can you quantify "sudden" here? Also, please elaborate how spectral changes in this case do not reflect the water content. What does this mean to the spectral changes in your Fig. 1 (and others) discussed above?**

**Reply:** Thank you for pointing this out. The word sudden was misplaced in this sentence. Here, we meant that not like sudden water content changes for inorganic aerosols, which coincides with spectral change, atmospheric aerosols can exhibit phase transition with gradual water content change along with spectral changes. Therefore, both aerosol water content change and spectral change should be analyzed to accurately depict evolutions of atmospheric aerosols. We have rephrased the sentence for clarification.

**Revision: Line 256** "Therefore, changes in water content and IR spectra of aerosols need to be considered collaboratively to accurately understand the reactions and phase transitions of these aerosols."

**Fig. 2: Please add labels for the red and pink shaded area directly into the Fig. to make inter-pretation easier.**

**Reply:** According to your advice, the labels have been put into the Fig. 3 (origin Fig.2).

**Revision:**

(a) SP:(NH$_4$)$_2$SO$_4$= 2:1 — ERH of Na$_2$SO$_4$, DRH of Na$_2$SO$_4$

(b) SP:NH$_4$Cl= 1:1 — ERH of SP

(c) SP:NH$_4$NO$_3$= 1:1 — ERH of NaNO$_3$, DRH of NaNO$_3$

—O— dehydration  —△— hydration  |—| sudden water loss  |—| sudden water uptake

**L224-241: • Please add references to appropriate subpanels in Fig. 2 directly to the text, as you jump quickly between the individual panels.**

**Reply:** Thank you. In the revised paper, the subpanels has been added when it was described and was also highlighted.

**Revision: line 256-257:** "Fig. 3 (Fig.2 in the original paper) shows the water content evolution during a RH cycle and their comparison with the phase transition point of compounds."

**• Please add information on the rate of RH changes during your experiments to the text. Was the rate slow enough to allow for equilibration of the particles and the surrounding RH?**

**Reply:** Thank you. This information has been added.

**Revision: line 253-254:** "The RH changes stepwise and the rate is < 5% min$^{-1}$. And the stay time at each level is 30 minutes to allow for equilibration between particles and the surrounding RH."

**L261: Check title, should this read: "The effect of molar ratio on the replacement…"**

**Reply:** Thank the careful check. The title has been revised.

**Revision: line 297:** "3.5 The effect of molar ratio on the replacement reaction and phase transition"

**L265: Please provide reasoning for the chosen ratio in the context of typical organic-to-inorganic ratio found in atmospheric aerosols.**

**Reply:** Thank you for the important suggestion. We added justification for the selected organic to inorganic ratio in Material and methods. In atmospheric aerosols, the groups

and ions were usually gained and it is difficult to measure detailed organic-to-inorganic ratio and chemical structure. However, it is well established that the emissions of various substances are dependent upon regions. Thus in this work, the ratios of 2:1, 1:1 and 1:2 represent different regions that may have excess organic matter, a balance between organic and inorganic matter, and excess inorganic matter.

**Revision: line 103-105:** "In the atmosphere, the organic to inorganic ratio varies regionally, but mostly remains at the same order of magnitude. Thus, the selected ratios herein can effectively represent different regions."

**L267-270: I suggest to repeat in parenthesis the meaning of the named peaks at e.g. 3130 cm-1 etc., to make it easier for the reader to follow here.**

**Reply:** The meaning of the 3130 cm-1 band has been added.

**Revision: line 303:** "…the 3130 cm$^{-1}$, $\nu$ (NH$_4^+$) band of solid NH$_4$Cl."

**L278: "From Fig. 4a…" Looking at the red squares in Fig. 4a, the shift from 1176/cm to 1186/cm appears to be pretty broad from about 40% RH to 20% RH. How can you get such a precise ERH value as 41% for the 1:1 mixture? Also, for the 2:1 mixture the peak shift seems to appear over an even broader RH range, so how do you get to the quoted 23.7% RH? Please clarify in the text.**

**Reply:** Thank you for your questions. In our experiment, the RH changes as stepwise mode rather than linear mode, and the IR spectra were measured at every RH levels. So the ERH value gained at a specific RH value. Even though the ERH range is broad, but the onset of efflorescence is a specific RH value. In the articles, 23.7% RH and 41% both refer to the onset of ERH rather than precise ERH, as described in line 315.

**Fig. 4: What is the difference between grey symbols with a horizontal vs. vertical grey line across the symbol?**

**Reply:** There are no difference and only a different symbol. In revised paper, the symbols have been corrected as grey circle.

[Figure]

**L294: Please delete "et al."**

**Reply:** Thanks. The "et al." has been deleted.

**Line 330-331:** "…replacement reaction and dissociation"

**L295: mixing → mixed, was → were**

**Reply:** Thank you, the "mixing" and "was" have been replaced by "mixed" and "were"

**Line 332:** "…various organic salts, mixed with $(NH_4)_2SO_4$ were measured…"

**L303: Replace "escalated" with more appropriate wording.**

**Reply:** The word "escalated" has been replaced by "became stronger".

**Line 340:** "…for crystalline sulfate became stronger"

**Fig. 7: Why was there no crystallization observed in the left-hand side images at 12.1% RH? Please clarify in text.**

**Reply:** The left-hand side images in Fig. 8 shows the phase evolution for 1:1 $ST/(NH_4)_2SO_4$ aerosols on dehydration, where no crystal was formed, as reflected by the absence of 995 $cm^{-1}$ band in IR spectra (Fig. 6(b)). So no crystallization was observed at 12.1%.

**L332-341: More information should be added how these experiments were done? What was the rate of RH changes? Was the first RH cycle started at high or low RH? How long were the particles exposed to these (high) RH conditions? The latter would help to get a better idea how fast these replacement reactions are.**

**Reply:** Thank you for your suggestions. The information of RH change has been added in revised paper.

**Revision: line 369-371:** "…the IR spectra of $SO_4^{2-}$ and organic salts remained distinguishable as the RH changes from ~88.7% to ~2.0% RH stepwise with the rate lower than 5% RH $min^{-1}$, indicating incomplete reaction likely due to mass transfer limitations in viscous aerosol particles for about 30 min at every RH level"

**L352: "substantial work" should be followed by appropriate refs.**

**Reply:** The refs have been added.

**Revision: line 390:** "previous substantial work on the phase state evolution of internally mixed organic/inorganic particles (Wang et al. 2019, Ma et al., 2022, Shao et al., 2018)."

**Line 601-603:** "Shao, X., Wu, F.-M., Yang, H., Pang, S.-F., Zhang, Y.-H.: Observing $HNO_3$ release dependent upon metal complexes in malonic acid/nitrate droplets, Spectrochim Acta A, 201, 399-404, https://doi.org/10.1016/j.saa.2018.05.026, 2018."

**L354: "retained in a viscous"**

**Reply:** Thanks, it has been corrected.

**Revision: line 393:** "SC particles retained in a viscous state at lower RH levels"

**L359: Please check if you really mean "dehydration" here or "hydration"?**

**Reply:** Yes, it is dehydration. In present work, SC and ST are both viscous state, but the phase behavior is different for their mixture with $(NH_4)_2SO_4$. For SC/$(NH_4)_2SO_4$ aerosols, crystalline solid $Na_2SO_4$ formed on dehydration, otherwise, it happened on hydration for ST/$(NH_4)_2SO_4$ particles.

**L360: "Grayson…" verb is missing. Please avoid repetition of references;**

**otherwise check for correct punctuation, i.e. "Grayson et al. have demonstrated..."**

**Reply:** Thanks. The sentence has been revised.

**Revision: line 399:** "Grayson et al. (2017) has confirmed viscosity increasing as the number of hydroxyl groups in the molecule."

**L361: "As the RH decreased…" Please add ref to this statement and provide a brief explanation of "gel" for the reader here (you could move the explanation you give on L367 up).**

**Reply:** In accordance with the reviewers' suggestions, we have provided a brief description of the gel.

**Revision: line 400:** "As the RH decreased, the polyhydroxy chain with a carboxyl group is prone to form a gel owing to intermolecular hydrogen bonding. Gels are two-phase mixtures of liquids dispersed in (semi-)solid amorphous matrices, and the uptake of water into a gel can involve gradual swelling as well as stepwise volume increases related to thermodynamically well-defined phase transitions (Pang et al., 2002)."

**L371: which → where**

**Reply:** Thanks, "which" has changed to "where"

**Revision: line 410:** "in these gels where mechanical entanglements"

**L372: How does this trapping work? Just by physical uptake of water due to capillary condensation or is this due to chemisorption? This should be added to the text.**

**Reply:** It is generally believed that the gel captures water due to both the chemical adsorption of water molecules onto the gel fibers and physical uptake of water due to capillary condensation. In revised version, the statement has been added.

**Revision: line 411-412:** "trap water molecules owing to both the chemical adsorption of water molecules onto the gel fibers and physical uptake of water due to capillary condensation"

**L373: Break up sentence: "… bound around fibres. Hence, migration of SO42- and Na+ ions was inhibited, so that these ions cannot come in contact and nucleate a crystalline phase."**

**Reply:** We thank the referee for the suggestion. The sentence has been corrected.

**Revision: line 412-414:** "Due to the strong interaction between OH groups and $SO_4^{2-}$ ions, anions were bound around fibers. Hence, migration of $SO_4^{2-}$ and $Na^+$ ions was inhibited, so that these ions cannot come in contact and nucleate a crystalline phase."

**Fig. 8: I like the idea of having a schematic, but I think this figure can be improved to clarify aspects described in the text. It is unclear what the different colors correspond to. A legend is missing to indicate the fibrous particles as "gel-like". Is there a better way to make the "OH groups and SO42- ions bound around the fibers" (L372) more obvious? Outgassing of NH3 is depicted but not mentioned in the text.**

**Reply: Thank you for your suggestion.** Fig. 8 has been improved by color-coding solid spheres to represent different ions. We hope it makes "ions bound around fibers" more obvious. Moreover, $NH_3$ release has been described in the article.

**Revision: Line 408:** "…gradually gelatinized water due to a decrease in water content, along with $NH_3$ release." And **line 417**: "…further form a solid nucleus, accompanying continuous $NH_3$ release."

[Figure]

**L387: "of atmospheric SOA." Please add more appropriate refs.**

**Reply:** Two refs have been added.

**Revision: Line 428-429:** "…of atmospheric secondary aerosols (Yang, et al., 2008; Kawamura et al., 2016; Huang et al., 2022)" and the refs are follows:

**Line 513-515:** "Kawamura, K., Bikkina, S.: A review of dicarboxylic acids and related compounds in atmospheric aerosols: Molecular distributions, sources and transformation. Atmos. Res., 170, 140−160, 10.1016/j.atmosres.2015.11.018, 2016."

**Line 637-638:** "Yang, L. M., Yu, L. E.: Measurements of oxalic acid, oxalates, malonic acid, and malonates in atmospheric particulates, Environ. Sci. Technol., 42 (24), 9268−9275, https://doi.org/10.1021/es801820z, 2008."

**L401-404: Please see my previous comment: Replacement reactions can certainly**

**impact the aerosol phase state as you document, but in turn these are also dependent on the phase state of the aerosol particle. This aspect should be better represented in your Conclusion section.**

**Reply:** Thank you. In revised paper, the impact of phase state in chemical process has been added.

**Revision: Line 445-448:** "Our findings highlight the intricate interplay between chemical components of organic/inorganic aerosol, such as organic molecular structure, molar ratio, replacement reaction and their collective impact on aerosol phase state, but in turn these are also dependent on the phase state of the aerosol particle. For example, the special chemical process in aerosols adopted gel state upon humidifying."

**L397-398: "Additionally, we observed…" This is certainly a very interesting finding of this study. I would like to see some more discussion, which atmospheric processes could be influenced by such "crystallization upon hydration". Also, can the authors speculate how important this process is and if they would expect it for other atmospherically relevant aerosol systems?**

**Reply:** Thank you for your comment. We very much appreciate your interest in "crystallization on hydration". Usually, the reactive uptake of gas from atmosphere occurs in liquid phase, which further provoke chemical composition evolution, and in turn change the optical property and CCN activity. This statement has been added in revised paper. We also expect similar phenomenon to happen in other dicarboxylic acids and AS mixture aerosols as long as it can potentially form gel-like structure prior to typical efflorescence during dehydration.

**Revision: Line 440-442:** "…which can reduce the possibility of further composition evolution and reactive gas uptake due to the absence of ion mobility in solid at higher RH. Hence, light absorbance and CCN activity, which strongly depend on chemical composition, will change. Future studies are needed to further investigate this phenomenon, as it is also expected to occur in other organic/inorganic aerosols that can potentially form gel-like state prior to typical efflorescence."

**L405-409: I am unclear what "targeted strategies to mitigate air pollution" the authors refer to. Please elaborate or remove this statement.**

**Reply:** According to suggestion, this statement has been deleted.

**Fig. 9: This figure is not mentioned and discussed in the text, please do that or delete the figure.**

**Reply:** Thanks, Figure 9 has been deleted.

---

## Author Response (AR1)

Guangjie Zheng, Ph.D.
Editor
*Atmospheric Chemistry and Physics*

Dear. Dr. Zheng:

My co-authors and I are pleased to submit a revision of the manuscript "The Interplay between Aqueous Replacement Reaction and the Phase State of Internally Mixed Organic/ammonium Aerosols". We note that the title of the paper has been revised according to referee's suggestion, and the title of the paper in the initial submission was "The Impact of Aqueous Phase Replacement Reaction on the Phase State of Internally Mixed Organic/ammonium Aerosols".

We are highly appreciative of the opportunity to revise this manuscript. In this revision, we have carefully responded to all the referee' questions, concerns, and recommendations by refining the language of the article, adding supporting literature and optimizing our discussion and data visualization. These suggestions are of great significance in enhancing the quality of our manuscript and facilitating readers' better understanding of our work.

In the response to referees, we list each comment by the reviewers. Followed by our detailed responses, corresponding revisions with accompanying line numbers, and additional figures. Specific changes are highlighted in yellow in the marked manuscript.

We sincerely hope that our work meets with your approval, and we greatly appreciate the time and effort that you and the referees have invested in our work.

Sincerely,
The authors

**Response to referee comments**

**Referee #1:**

In Yang and co-workers describe laboratory measurements of internally mixed organic-inorganic aerosol systems. Mainly based on FTIR, their experiments demonstrate how replacement reactions in aqueous aerosols can alter the composition, with subsequent implications for the aerosol phase state, an important property that determines the impacts of aerosol particles on air quality and climate.

Overall, I find the experiments interesting and the topic falls within the scope of Atmospheric Chemistry and Physics. However, I would like the authors to address the points below, before their work can be considered for publication.

**Author reply:** We are grateful for the careful review, the positive feedback, and the constructive suggestions from the referee. These suggestions are of great significance in enhancing the quality of our manuscript and facilitating readers' better understanding of our work. We have incorporated the referee's comments and suggestions into our manuscript. In the response to the referee, we list each comment, followed by our detailed responses, corresponding revisions with accompanying line numbers, and additional figures. Specific changes are noted in the red in the marked manuscript. We believe that the changes have resulted in a much stronger manuscript.

**1L16: Please rewrite and clarify the list of numbers: Do you mean 65.5% to 60.1%?**

**Reply:** Thanks for the careful review. We have carefully checked the manuscript and corrected this error.

**Revision: line 15-17:** "For SP/ammonium aerosols, $NaNO_3$ and $Na_2SO_4$ crystallized from 35.7% to 12.7%, and from 65.7% to 60.1% RH, respectively, lower than pure inorganics (62.5±9−32% RH for $NaNO_3$ and 82±7−68±5% RH for $Na_2SO_4$)."

**L17: Please add RH uncertainties to your quoted DRH values. Consider also adding the DRH values of pure inorganics in parenthesis at the end of this sentence, so that the increase in DRH becomes more quantitative.**

**Reply:** Thank you for this helpful suggestion. We have added the uncertainties and the DRH of pure $Na_2SO_4$ and $NaNO_3$, which come from other's work.

**Revision: line 17-19:** "Upon hydration, the crystalline $Na_2SO_4$ and $NaNO_3$ deliquesced at 88.8%−95.2% and 76.5±2−81.9%, higher than those of pure $Na_2SO_4$ (74±4%−98% RH) and $NaNO_3$ (65−77.1±3% RH)."

**L29: Please check your reference formatting. This reference is from the year 2011, not 2001.**

**Reply:** Thank you for pointing this out. Because the introduction has been rewritten according to the second suggestion and this reference has been deleted.

**L33: Please check your reference formatting. This reference should read**

**"Kreidenweis".**

**Reply:** Thank you for pointing this out. We have corrected the reference format and checked the format for other for our manuscript.

**Revision: line 30:** "…cloud condensation nuclei (CCN) activity (Kreidenweis et al., 2005),"

**L37: "… with carboxylic acids/salts and ammonium salts as one of the most abundant…" Please add appropriate references to this statement.**

**Reply:** This sentence has been revised and references has been added.

**Revision: line 36-37:** "Atmospheric aerosols are often composed of inorganic sulfates, nitrates, and organic species in varying mixing ratios (Riemer et al., 2018; Huang et al., 2014; Trebs et al., 2005; Zhou et al., 2019)"

**Line 514-518: "**Huang, R.-J., Zhang, Y., Bozzetti1, C., Ho, K.-F., Cao, J.-J., Han, Y., Daellenbach, K. R., Slowik, J. G., Platt, S. M., Canonaco, F., Zotter, P., Wolf, R., Pieber, S. M., Bruns, E. A., Crippa, M., Ciarelli, G., Piazzalunga, A., Schwikowski, M., Abbaszade, G., Schnelle-Kreis, J., Zimmermann, R., An, Z., Szidat, S., Baltensperger, U., Haddad, I. E., Prévôt, A. S. H.: High secondary aerosol contribution to particulate pollution during haze events in China, Nature, 514, 218-222, http:// doi:10.1038/nature13774, 2014."

**line 601-602:** "Riemer, N., Ault, A. P., West, M., Craig, R. L., Curtis, J. H.: Aerosol Mixing State: Measurements, Modeling, and Impacts, Rev. Geophys. 57, 187−249, https://doi.org /10.1029/2018RG000615, 2019."

**line 623-626:** Trebs, I.; Metzger, S.; Meixner, F. X.; Helas, G. N.; Hoffer, A.; Rudich, Y.; Falkovich, A. H.; Moura, M. A. L.; da Silva, R. S.; Artaxo, P.; Slanina, J.; Andreae, M. O. The $NH_4^+$-$NO_3^-$-$Cl^-$-$SO_4^{2-}$-$H_2O$ aerosol system and its gas phase precursors at a pasture site in the amazon basin: How relevant are mineral cations and soluble organic acids? J. Geophys. Res.-Atmos. 110 (D7), D07303, http://doi.org/10.1029/2004JD005478, 2005.

**L44: "Organics can decrease…". Please add appropriate references to this statement, as well as to the following sentence.**

**Reply:** Thank you very much for your suggestion. The abstract has been rewritten and the sentence has been deleted.

**L47: Please add more relevant references, e.g.: 10.1021/jp0556759**

**Reply:** Thank you. The sentence has been written and reference has been added in revised paper.

**Revision: line 37-39:** "The interaction between organic and inorganic components can alter aerosol phase state and hygroscopicity (Shi et al., 2017; Wang et al., 2017; Marcolli et al., 2006)."

line 566-568: "Marcolli, C., Kreidenweis, U. K.: Phase Changes during Hygroscopic Cycles of Mixed Organic/Inorganic Model Systems of Tropospheric Aerosols, J. Phys. Chem. A, 110, 1881–1893, https://doi.org/10.1021/jp0556759, 2006."

**L50-51: Please either give a review article here as reference, or add further primary refs for each of the named processes, i.e. reactivity, heterogeneous uptake…**

Reply: Thank you for your suggestion, the sentence has been rewritten and we cited a review article as our reference.

Revision: line 50: "Aerosol phase state significantly influences chemical reaction within aerosols (Herrmann et al., 2015)."

Line 508-510: "Herrmann, H., Schaefer, T., Tilgner, A., Styler, S. A., Weller, C., Teich, M.; Otto, T.: Tropospheric Aqueous-Phase Chemistry: Kinetics, Mechanisms, and Its Coupling to a Changing Gas Phase, Chem. Rev. 115, 4259-4334, https://doi.org/10.1021/cr500447k, 2015."

**L52: This is the first time you mention "replacement reaction". I would encourage the authors to give a brief, general definition and maybe a relevant example, so that a reader can follow more easily.**

Reply: We thank the referee for the suggestion. The definition and an example of replacement reaction has been added in revised paper and highlighted.

Revision: line 53-56: "In aqueous aerosols, the ions between two ionic compounds can exchange to form new compounds through aqueous replacement reactions. For instance, when oxalic acid mixes with nitrates, the $H^+$ ions can combine with $NO_3^-$ ions to form $HNO_3$, which off-gases, leading to the formation of metal oxalate salts in the aerosol (Ma et al. 2019)."

**L54: Check reference: "Ma et al. (Ma et al., 2019)"**

Reply: Thanks, it has been corrected.

Revision: line 56: "…leading to the formation of metal oxalate salts in the aerosol (Ma et al. 2019)."

**L56: "… the impact of … on hygroscopic growth": It would be good to more precisely specify the impact. Did oxalate formation lead to a decrease or increase in hygroscopic growth?**

Reply: Thank you for the comment. This sentence has been revised in rewritten introduction.

Revision: line 58-60: "…These reactions can generate gaseous compounds or less hygroscopic compounds, causing the substitution of weak bases for strong bases in aerosols (Yang et al 2019; Du et al 2020; 2021; Chen et al, 2022)"

**L57: Check reference: "Ma et al. (Ma et al., 2022)"**

**Reply:** I have checked the ref. and corrected the reference.

**Revision: line 51:** "Phase changes of aerosol components can enhance chemical reactions (Ma et al., 2022…)"

**L60: Consider changing "liberate" to "release" or "formation"**

**Reply:** Thank you. In rewritten introduction, this sentence has been deleted.

**L64: No need to write Chen et al. twice in the sentence. Rewrite as : "Chen et al. (2020) elucidated…., respectively." Also, is it really appropriate to say "elucidated", since in the next sentence you write how information on aerosol phase state was missing.**

**Reply:** Thanks, the reference has been corrected. The sentence of "Chen et al. (2020) elucidated…., respectively." has been deleted because the introduction has been rewritten.

**L64: I am unclear what "lack of information" or "uncertainty in aerosol phase state" you refer to, please specify? Do you mean a lack of knowing the crystallization RH?**

**Reply:** Thank you for your comment. The sentence has been deleted in revised text.

**L70-72: This sentence reads cumbersome, please reformulate. I think you are trying to say that phase states such as LLPS can promote the inhomogeneity of components? Your references seems appropriate for the first part ("The inhomogeneity of aerosol components can significantly impact the degree of reactivity"). I would move it there and then consider adding other references for the occurrence of LLPS, liquid-solid coexistence etc.**

**Reply:** Thank you for the suggestions. The sentence has been revised for clarity and a reference was also added.

**Revision: line 63-66:** "The complexity of atmospheric aerosol phase states - including the coexistence of crystalline solid and liquid (Yang et al., 2019), liquid-liquid phase separation (Zhou et al., 2019), and solid in a viscous phase (Zhu et al., 2022) - can lead to inhomogeneity of aerosol components, which in turn affects their reactivity. (Zong et al., 2022)."

**Line 662-664:** "Zhou, S., Hwang, B. C. H., Lakey, P. S. J., Zuend, A., Abbatt, J. P. D., Shiraiwa, M.: Multiphase reactivity of polycyclic aromatic hydrocarbons is driven by phase separation and diffusion limitations, 24, 11658-11633, https://doi.org/10.1073/pnas.1902517116, 2019."

**L76-79: "Therefore, …" I am a bit stuck with this statement. On L69-70 you write that "replacement reactions and phase state are intercorrelated parameters", which I think is a fair way to put it. Now, on this line here you argue about the impact of replacement reactions *on* aerosol phase state. However, at the same time these replacement reactions to in turn depend on the aerosol phase state. What is**

**the correct line of reasoning, i.e. replacement reactions affecting phase state or phase state affecting replacement reactions…? Maybe it would be fairer to reformulate this here to something like: "Exploring the interplay of different organic and inorganic salts on reactivity, particularly the reactivity of replacement reactions, on the one hand side, and aerosol phase state on the other hand side, is important to understand atmospheric aerosols". The title of the manuscript should be adjusted accordingly.**

**Reply:** Thank you very much. The sentence has been rewritten as your recommendation. And the title of the manuscript has been adjusted.

**Revision: line 69-71:** "Therefore, investigating the interplay between different organic and inorganic salts in terms of their reactivity and aerosol phase state is crucial for understanding atmospheric aerosols."

**Line 1-3:** "The Interplay between Aqueous Replacement Reaction and the Phase State of Internally Mixed Organic/ammonium Aerosols"

**L77: Change "significance" to "important"**

**Reply:** Thank you. The sentence has been rewritten and the "crucial" was used.

**Line 70:** "…in terms of their reactivity and aerosol phase state is crucial for understanding…"

**L87: See my comment above: It might be good to replace "correlation" here with "interplay"**

**Reply:** Thanks, we have replaced correlation with "interplay".

**Revision: line 80:** "This study improves our understanding of the interplay between aerosol composition…"

**L93: What is "three distilled water"?**

**Reply:** Thank you. The "three distilled water" means that the water was distilled three times. It has been corrected as "triple distilled water"

**Revision: line 85:** "…were prepared using triple distilled water."

**L97: Please indicate compound purities when introducing the chemicals, along with supplier.**

**Reply:** Thanks for your suggestion. The purity and supplier has been added in revised paper.

**Revision: line 86-89:** "The organic acid salts include sodium pyruvate (SP, Aladdin Reagent Co., Ltd., ≥99.0%), sodium citrate (SC, Beijing Chemical Reagents Company, ≥99.0%), and sodium tartrate (ST, Beijing Chemical Reagents Company, ≥99.0%). The ammonium salts include $(NH_4)_2SO_4$, $NH_4NO_3$ and $NH_4Cl$ (all from Beijing Chemical Reagents Company, ≥99.0%)."

**L97: How was this diameter measured? Please specify. Also, were FTIR experiments done for differently sized particles? How was particle homogeneity verified, i.e. absence of e.g. liquid-liquid phase separation, that could affect results.**

**Reply:** We thank the reviewer for the comment. The dimeter was measured by optical microscopy. In our experiment, droplets were deposited onto the substrate and kept at the highest level (>87% RH) for about 30 min before size measurement. The droplet size distribution follows a Gaussian distribution with a mean of 5 micrometers. The homogeneity of the droplets was also verified according to optical images. No LLPS was observed at relatively high RH prior to efflorescence.

**Table 1: Please add spaces between numeric values and units, e.g. 100 g instead of 100g.**

**Reply:** Thanks, we have checked the whole paper and add spaces between numeric values and units. The revised parts have been highlighted.

**Revision:**

**Table 1: The molecular structure of inorganic and organic compounds**

| Compound | Molecular structure | Dissociation constant of conjugate acid | Solubility in 100g H$_2$O (20°C) | Molecular weight (Da) |
|---|---|---|---|---|
| Ammonium nitrate | NH$_4$NO$_3$ | **NA** | 190 g | 80.043 |
| Ammonium chloride | NH$_4$Cl | **NA** | 37.2 g | 53.49 |
| Ammonium sulfate | (NH$_4$)$_2$SO$_4$ | **NA** | 75.4 g | 132.14 |
| Sodium pyruvate |  | $K_a = 3.2 \times 10^{-3}$ | 47 g | 82.03 |
| Sodium citrate |  | $K_{a,1} = 7.4 \times 10^{-4}$ $K_{a,2} = 1.7 \times 10^{-5}$ $K_{a,3} = 4.0 \times 10^{-7}$ | 154 g | 258.07 |
| Sodium tartrate |  | $K_{a,1} = 1.04 \times 10^{-3}$ $K_{a,2} = 4.55 \times 10^{-5}$ | 33.3 g | 194.05 |

**L104: bump → pump**

**Reply:** The "bump" has been replaced by "pump".

**Revision: line 99:** "…mass flowmeter and vacuum pump…"

**Section 3 general: It would be helpful to add a Table to the SI where the different peaks and the corresponding chemical groups and inferred phase state were summarized, as it is in parts very hard to follow the discussion throughout this Section.**

**Reply:** We thank the reviewer for the helpful suggestion. Table S1 has been added, as well as the statement in paper.

**Revision: line 136-137:** "…and band assignments can be found in Table S1"

**Table S1** Assignments of IR Peaks Attributed to Ammonium and Carboxylates

| | | | | | | Assignment |
|---|---|---|---|---|---|---|
| | | | 831 | | | $\nu_2$-$NO_3^-$ |
| | | 1055(s) | | | | $\gamma(CH_2)$ |
| | | 1069(l) | | | | $\gamma(CH_2)$ |
| 1176(l) | | | | | | $\nu$ (C-C) + $\gamma$ (COO$^-$)(liquid) |
| | | | | | 1180 | |
| 1186(s) | | | | | | $\nu_3(SO_4^{2-})$ |
| | 1278 | 1276(s) | | | | $\nu$ (C-C) + $\gamma$ (COO$^-$) |
| | 1308 | | | | | $\nu$(C-C) |
| 1353 | | | | | | $\omega(CH_2)$ |
| | 1386 | 1388(l) | | | | $\delta_s$ (CH$_3$) |
| 1404(s) | | | 1417(s) | 402(s) | 1412(s) | $\nu_s$ (COO$^-$) |
| | | | | | | $\nu_2(NH_4^+)$(solid) |
| 1424(l) | | 1436(s) | 1448(l) | 1440(l) | 1443(l) | $\nu_s$ (COO$^-$) |
| | | | | | | $\nu_2(NH_4^+)$(liquid) |
| | 1568 | | | | | $\nu_{as}$ (COO$^-$) |
| 1608(l) | | | | | | $\delta$ (OH) + $\nu$ (CO) |
| 1627(s) | | | | | | $\nu_s$ (COO$^-$) |
| 1654(s) | | | | | | $\nu_s$ (COO$^-$) |
| 1709 | | | | | | $\nu_s$ (COO$^-$) |
| | | | | | | $\nu_{as}$ (COO$^-$) |
| | | | | | | $\nu_{as}$ (C=O) |

| | | |
|---|---|---|
| | 1754 | |
| | 2804 | $\nu$ (NH$_4^+$) |
| | 3036 | $\nu$ (NH$_4^+$) |
| 3360 | 3130 | $\nu$ (NH$_4^+$) |
| | | $\nu_{as}$(O–H) |

s: solid state,  l: liquid state

**L126: add "is" shifted to**

**Reply:** Thank for your careful reading, the "is" has been added.

**Revision: line 141:** "1176 cm$^{-1}$ is shifted to 1186 cm$^{-1}$"

**L140: I would encourage to specify here and elsewhere as "crystalline solid phase", as the "viscous phase" can also be solid (== glassy) or semi-solid, which both denote amorphous solid phase states.**

**Reply:** The advice is very good. We have specified the "crystalline solid phase" in appropriate place to avoid the confusion with amorphous solid phase states, all of which have been highlighted.

**L142: Here and elsewhere, please give RH uncertainties.**

**Reply:** In revised paper, the RH uncertainties have been added in some places and highlighted.

**Revision: line 159:** "…where an error margin is 1%."

**L145-147: I am unclear about what you write here. If the sodium citrate aerosol is highly viscous, why is the water uptake than not gradual, but "abrupt", as you write? To me the humidification curves on sodium citrate and sodium tartrate look like curves with a sharp deliquescence, which I would only expect for crystalline material. Please clarify why you nonetheless think that these phase states are "viscous", which you seem to use to describe an amorphous solid state.**

**Reply:** The viscous state was concluded by little water retained at the lowest RH, which can be seen from the water content dependent upon the RH during dehydration as seen in Fig. 1. The more detailed description has been added. On hydration, the viscous state can transform into solution at a certain RH, which is gel-broken point.

**Revision: line 154-155:** "…and the retained normal water content of 0.25 and 0.15 for ST and SC, respectively, at the lowest RH…"

**Section 3: This Section has many different subsections and the connection of these is not always obvious. It would be good to add a brief description of what topic is treated where to guide the reader a bit at the start of the Results and Discussions.**

**Reply:** Based on referee's suggestion. A brief description that guides the reader to

different topics was added at the start of the Results and Discussions.

**Revision: line 115-133:** "The phase behaviour of mixed aerosols is dependent upon the chemical composition, molar ratio and chemical process. Herein we selected three organic salts including SP, ST and SC, along with $NH_4Cl$, $NH_4NO_3$ and $(NH_4)_2SO_4$, to study the interplay between phase state, composition evolution, and aqueous phase replacement reactions in aerosols during RH changing cycles. This section consists of seven parts. In *part 3.1*, we analysed the IR spectra of pure organic salts and inorganic salts to obtain the differences in characteristic peaks for aqueous and solid phases. Based on this, in *part 3.2*, SP was mixed with different ammonium salts with various stoichiometric ratio to form aerosols, and the infrared spectra of the mixed aerosols were tested to analyse the characteristic absorption peaks in the second section. The results showed that SP underwent a substitution reaction with the ammonium salts, and the mixed aerosols containing different ammonium salts exhibited distinct phase behaviours. Therefore, in *parts 3.3 and 3.4*, we discussed in detail the substitution reactions and the resulting changes in water content and phase transition regions of the compounds. The phase transition regions of the reaction products were compared with those of the pure components. The relative content of the mixed components may vary in different regions. In *part 3.5*, we took the SP/NH4Cl system as an example to study the impact of different component contents on phase transition behaviour. After investigating the hygroscopic behaviour of mixed aerosols containing different ammonium salts with the same organic salt, in *part 3.6*, we examined the phase evolution of aerosols induced by mixing different organic salts, SC and ST, with ammonium sulphate. By combining infrared spectroscopy and optical images, we discovered that the mixture of tartaric acid and ammonium sulphate not only underwent substitution reactions but also exhibited unexpected hygroscopic weathering behaviour. Therefore, in *part 3.7*, we discussed the causes of this special phase behaviour."

**Section 3.1 general: It could make sense to move your Fig. S2 to the main text. That way, a reader could follow the described spectral changes with changes in RH better. Also, Fig. S2 misses a proper legend indicating what the black and red data points correspond to.**

**Reply:** Thanks for the advice. Fig. S2 have been moved to the main text as Fig. 1 and a proper legend has been added.

[Figure]

Fig. 1. The IR spectra of organic salts on dehydration and hygroscopic behavior during a down-up RH cycle. The shaded area shows the chosen integration region for liquid water. The spectra for sodium pyruvate was previous reported by Yang et al (2019).

**L153-155/Fig.1: In the blue-framed enlarged image: Where is the blue and the red lines? Please color your lines as in the left most panel or the violet-framed enlarged image. Otherwise, it is impossible to connect the RH values to the individual lines. Note, this also concerns Fig. 1c.**

**Reply:** Thanks for the advice. The lines in the left panel and right enlarged curves are correspondingly matched in the revised Fig. 1.

**Revision:**

[Figure]

**L159: Reference formatting: No need to have "Tan et al." twice in this sentence, please change.**

**Reply:** Thank you. This has been corrected in revised paper.

**Revision: line 179:** "…RH. Tan et al (2014) investigated…"

**L185-195: Please add appropriate references the subpanels of Fig. S3, to make it clearer what you are describing here.**

**Reply:** Thank you. In revised version, a sentence to describe Fig. S2 (Fig. S3 in the original SI) has been added.

**Revision: line 209-210:** "Following the dehydration, the hydration of three SP/ammonium aerosols was performed. Fig. S2 presents the IR spectra on hydration."

**Section 3.3 general: The structure of this section should be improved. In the beginning, the authors introduce relevant replacement reactions. The middle part (L202-210 then discusses some literature findings. The last part (L210-235) then discusses hygroscopicity data from the present manuscript, but the link to the topic of replacement reactions remains unclear. This requires improvement in a revised version.**

**Reply:** Thank you for your suggestion. We believe that the whole paragraph has been improved following your suggestion. We have revised Section 3.3 including the topic and the first paragraph.

**Revision: line 224-238:** " **3.3 The replacement reactions in SP/ammonium aerosols and the resulting hygroscopicity**

Above analysis on IR features showed the following reactions (1) - (3) in mixed SP/ammonium aerosols:

$$CH_3COCOONa(aq) + (NH_4)_2SO_4(aq) \xrightarrow{RH\ decreasing} CH_3COCOOH(g) + NH_3(g) + Na_2SO_4(s) \quad (1)$$

$$CH_3COCOONa(aq) + NH_4Cl(aq) \xrightarrow{RH\ decreasing} CH_3COCOOH(g) + NH_3(g) + NaCl(s) \quad (2)$$

$$CH_3COCOONa(aq) + NH_4NO_3(aq) \xrightarrow{RH\ decreasing} CH_3COCOOH(g) + NH_3(g) + NaNO_3(s) \quad (3)$$

It can be seen that pyruvic acid and ammonia are formed and depleted from particles alongside the crystalline solid formation of various inorganic salts during dehydration, which proceeded the replacement reaction between SP and ammonium. Herein, aerosol particles act as micro-reactors, with their larger specific surface area than bulk solution facilitating similar processes. In fact, the replacement reactions in aerosols driven by gas released or compound formation with lower solubility have been reported in previous work. For example, Wang et al. (2017) observed the reaction in aerosols composed of oxalic acid and ammonium sulfate, which was driven by formation of lower hygroscopic ammonium hydrogen oxalate ($NH_4HC_2O_4$) and ammonium hydrogen sulfate ($NH_4HSO_4$) during the dehydration process. While Wang et al. (2019) addressed the fact of the formation of crystalline solid $Na_2SO_4$ from $(CH_2)_n(COONa)_2$ (n = 1, 2)/$(NH_4)_2SO_4$ aerosols upon dehydration. Building upon their findings, crystalline solid $NaNO_3$ and $NaCl$ are also formed, as shown in equations (2) and (3)."

**L198-200: Please indicate phase states of reactants too. Please also indicate whether these reactions take place for decreasing or increasing RH.**

**Reply:** Thank you for the suggestion. The phase states of reactants and reaction condition have been added in revised paper.

**Revision: line 226-228:**

$$CH_3COCOONa(aq) + (NH_4)_2SO_4(aq) \xrightarrow{RH\ decreasing} CH_3COCOOH(g) + NH_3(g) + Na_2SO_4(s) \quad (1)$$

$$CH_3COCOONa(aq) + NH_4Cl(aq) \xrightarrow{RH\ decreasing} CH_3COCOOH(g) + NH_3(g) + NaCl(s) \quad (2)$$

$$CH_3COCOONa(aq) + NH_4NO_3(aq) \xrightarrow{RH\ decreasing} CH_3COCOOH(g) + NH_3(g) + NaNO_3(s) \quad (3)$$

**L206: "larger specific surface area" compared to what? Bulk solutions? Unclear.**

**Reply:** The aerosol particles are microdroplets, which has larger specific surface area than bulk solution under the condition of the same volume. In revised paper, the

sentence has been corrected.

**Revision: line 228:** "…with their larger specific surface area than bulk solution facilitating…"

**L215: "…water loss or uptake". Add https://doi.org/10.1021/cr990034t or other appropriate refs.**

**Reply:** Thanks. the literature has been added in revised paper and highlighted.

**Revision: line 240:** "…water loss or uptake (Martin, 2000)"

**Line 569-570** "Martin, S. T.: Phase Transitions of Aqueous Atmospheric Particles, Chem. Rev., 100, 3403–3453, https://doi.org/10.1021/cr990034t, 2000."

**L221: "These processes…": Can you quantify "sudden" here? Also, please elaborate how spectral changes in this case do not reflect the water content. What does this mean to the spectral changes in your Fig. 1 (and others) discussed above?**

**Reply:** Thank you for pointing this out. The word sudden was misplaced in this sentence. Here, we meant that not like sudden water content changes for inorganic aerosols, which coincides with spectral change, atmospheric aerosols can exhibit phase transition with gradual water content change along with spectral changes. Therefore, both aerosol water content change and spectral change should be analyzed to accurately depict evolutions of atmospheric aerosols. We have rephrased the sentence for clarification.

**Revision: Line 247** "Therefore, changes in water content …"

**Fig. 2: Please add labels for the red and pink shaded area directly into the Fig. to make inter-pretation easier.**

**Reply:** According to your advice, the labels have been put into the Fig. 3 (origin Fig.2).

**Revision:**

[Figure]

**L224-241: Please add references to appropriate subpanels in Fig. 2 directly to the text, as you jump quickly between the individual panels.**

**Reply:** Thank you. In the revised paper, the subpanels have been added when it was described and was also highlighted.

**Revision: line 249-250:** "Fig. 3 (Fig.2 in the original paper) shows the water content evolution during a RH cycle and their comparison with the phase transition point of compounds."

**• Please add information on the rate of RH changes during your experiments to the text. Was the rate slow enough to allow for equilibration of the particles and the surrounding RH?**

**Reply:** Thank you. This information has been added.

**Revision: line 250-251:** "The RH changes stepwise and the rate is $< 5\%$ $min^{-1}$. And the stay time at each level is 30 minutes to allow for equilibration between particles and the surrounding RH."

**L261: Check title, should this read: "The effect of molar ratio on the replacement…"**

**Reply:** Thank the careful check. The title has been revised.

**Revision: line 290:** "3.5 The effect of molar ratio on the replacement reaction and phase transition"

**L265: Please provide reasoning for the chosen ratio in the context of typical organic-to-inorganic ratio found in atmospheric aerosols.**

**Reply:** Thank you for the important suggestion. We added justification for the selected organic to inorganic ratio in Material and methods. In atmospheric aerosols, the groups and ions were usually gained and it is difficult to measure detailed organic-to-inorganic ratio and chemical structure. However, it is well established that the emissions of various substances are dependent upon regions. Thus in this work, the ratios of 2:1, 1:1 and 1:2 represent different regions that may have excess organic matter, a balance between organic and inorganic matter, and excess inorganic matter.

**Revision: line 76-78:** "The mole ratio of 1:2, 1:1 and 2:1 was applied to simulate the sulplus organic salts, equal and excessive inorganic slats for mixed aerosols depending on location, aerosol source and season." And **line 90-92:** "In the atmosphere, the organic to inorganic ratio varies regionally, but mostly remains at the same order of magnitude. Thus, the selected ratios herein can effectively represent different regions."

**L267-270: I suggest to repeat in parenthesis the meaning of the named peaks at e.g. 3130 cm-1 etc., to make it easier for the reader to follow here.**

**Reply:** The meaning of the 3130 cm-1 band has been added.

**Revision: line 296:** "…the 3130 $cm^{-1}$, $\nu$ ($NH_4^+$) band of solid $NH_4Cl$."

**L278: "From Fig. 4a…"** Looking at the red squares in Fig. 4a, the shift from 1176/cm to 1186/cm appears to be pretty broad from about 40% RH to 20% RH. How can you get such a precise ERH value as 41% for the 1:1 mixture? Also, for the 2:1 mixture the peak shift seems to appear over an even broader RH range, so how do you get to the quoted 23.7% RH? Please clarify in the text.

**Reply:** Thank you for your questions. In our experiment, the RH changes as stepwise mode rather than linear mode, and the IR spectra were measured at every RH levels. So the ERH value gained at a specific RH value. Even though the ERH range is broad, but the onset of efflorescence is a specific RH value. In the articles, 23.7% RH and 41% both refer to the onset of ERH rather than precise ERH, as described in **line 308-309**.

**Fig. 4: What is the difference between grey symbols with a horizontal vs. vertical grey line across the symbol?**

**Reply:** There is no difference but only a different symbol. In revised paper, the symbols have been corrected as grey circle.

[Figure]

[Figure]

**L294: Please delete "et al."**

**Reply:** Thanks. The "et al." has been deleted.

**Line 323-324:** "…replacement reaction and dissociation"

**L295: mixing → mixed, was → were**

**Reply:** Thank you, the "mixing" and "was" have been replaced by "mixed" and "were"

**Line 325:** "…various organic salts, mixed with $(NH_4)_2SO_4$ were measured…"

**L303: Replace "escalated" with more appropriate wording.**

**Reply:** The word "escalated" has been replaced by "became stronger".

**Line 333:** "…for crystalline sulfate became stronger"

**Fig. 7: Why was there no crystallization observed in the left-hand side images at 12.1% RH? Please clarify in text.**

**Reply:** The left-hand side images in Fig. 8 shows the phase evolution for 1:1 ST/(NH$_4$)$_2$SO$_4$ aerosols on dehydration, where no crystal was formed, as reflected by the absence of 995 cm$^{-1}$ band in IR spectra (Fig. 6(b)). So no crystallization was observed at 12.1%. In text, it has been added in line 352-354.

**Line 353-355:** "…As shown in Fig. 8, The left-hand side images shows the s round and smooth particle on dehydration, indicative of a homogeneous aqueous state without crystal formed till 12.1%, which is consistent with the absence of 996 cm$^{-1}$ band in IR spectra (Fig. 6(b))"

**L332-341: More information should be added how these experiments were done? What was the rate of RH changes? Was the first RH cycle started at high or low RH? How long were the particles exposed to these (high) RH conditions? The latter would help to get a better idea how fast these replacement reactions are.**

**Reply:** Thank you for your suggestions. The information of RH change has been added in revised paper.

**Revision: line 365-367:** "…the IR spectra of SO$_4$$^{2-}$ and organic salts remained distinguishable as the RH changes from ~88.7% to ~2.0% RH stepwise with the rate lower than 5% RH min$^{-1}$, indicating incomplete reaction likely due to mass transfer limitations in viscous aerosol particles for about 30 min at every RH level"

**L352: "substantial work" should be followed by appropriate refs.**

**Reply:** The refs have been added.

**Revision: line 387:** "previous substantial work on the phase state evolution of internally mixed organic/inorganic particles (Wang et al. 2019, Ma et al., 2022, Shao et al., 2018)."

**Line 606-608:** "Shao, X., Wu, F.-M., Yang, H., Pang, S.-F., Zhang, Y.-H.: Observing HNO$_3$ release dependent upon metal complexes in malonic acid/nitrate droplets, Spectrochim Acta A, 201, 399-404, https://doi.org/10.1016/j.saa.2018.05.026, 2018."

**L354: "retained in a viscous"**

**Reply:** Thanks, it has been corrected.

**Revision: line 389:** "SC particles retained in a viscous state at lower RH levels"

**L359: Please check if you really mean "dehydration" here or "hydration"?**

**Reply:** Yes, it is dehydration. In present work, SC and ST are both viscous state, but the phase behavior is different for their mixture with (NH$_4$)$_2$SO$_4$. For SC/(NH$_4$)$_2$SO$_4$ aerosols, crystalline solid Na$_2$SO$_4$ formed on dehydration, otherwise, it happened on hydration for ST/(NH$_4$)$_2$SO$_4$ particles.

**L360: "Grayson…" verb is missing. Please avoid repetition of references;**

**otherwise check for correct punctuation, i.e. "Grayson et al. have demonstrated…"**

**Reply:** Thanks. The sentence has been revised.

**Revision: line 395-396:** "Grayson et al. (2017) has confirmed viscosity increasing as the number of hydroxyl groups in the molecule."

**L361: "As the RH decreased…" Please add ref to this statement and provide a brief explanation of "gel" for the reader here (you could move the explanation you give on L367 up).**

**Reply:** In accordance with the reviewers' suggestions, we have provided a brief description of the gel.

**Revision: line 397-399:** "Gels are two-phase mixtures of liquids dispersed in (semi-)solid amorphous matrices, and the uptake of water into a gel can involve gradual swelling as well as stepwise volume increases related to thermodynamically well-defined phase transitions (Pang et al., 2002)."

**L371: which ➔ where**

**Reply:** Thanks, "which" has changed to "where"

**Revision: line 406:** "in these gels where mechanical entanglements"

**L372: How does this trapping work? Just by physical uptake of water due to capillary condensation or is this due to chemisorption? This should be added to the text.**

**Reply:** It is generally believed that the gel captures water due to both the chemical adsorption of water molecules onto the gel fibers and physical uptake of water due to capillary condensation. In revised version, the statement has been added.

**Revision: line 407-408:** "…trap water molecules owing to both the chemical adsorption of water molecules onto the gel fibers and physical uptake of water due to capillary condensation"

**L373: Break up sentence: "… bound around fibres. Hence, migration of SO42- and Na+ ions was inhibited, so that these ions cannot come in contact and nucleate a crystalline phase."**

**Reply:** We thank the referee for the suggestion. The sentence has been corrected.

**Revision: line 408-410:** "Due to the strong interaction between OH groups and $SO_4^{2-}$ ions, anions were bound around fibers. Hence, migration of $SO_4^{2-}$ and $Na^+$ ions was inhibited, so that these ions cannot come in contact and nucleate a crystalline phase."

**Fig. 8: I like the idea of having a schematic, but I think this figure can be improved to clarify aspects described in the text. It is unclear what the different colors correspond to. A legend is missing to indicate the fibrous particles as "gel-like". Is there a better way to make the "OH groups and SO42- ions bound around the fibers" (L372) more obvious? Outgassing of NH3 is depicted but not mentioned in the text.**

**Reply: Thank you for your suggestion.** Fig. 8 has been improved by color-coding solid spheres to represent different ions. We hope it makes "ions bound around fibers" more obvious. Moreover, $NH_3$ release has been described in the article.

**Revision: Line 404:** "…gradually gelatinized water due to a decrease in water content, along with $NH_3$ release." And **line 413**: "…further form a solid nucleus, accompanying continuous $NH_3$ release."

[Figure]

**L387: "of atmospheric SOA." Please add more appropriate refs.**

**Reply:** Two refs have been added.

**Revision: Line 425-426:** "…of atmospheric secondary aerosols (Yang, et al., 2008; Kawamura et al., 2016; Huang et al., 2022)" and the refs are follows:

**Line 526-528:** "Kawamura, K., Bikkina, S.: A review of dicarboxylic acids and related compounds in atmospheric aerosols: Molecular distributions, sources and transformation. Atmos. Res., 170, 140−160, 10.1016/j.atmosres.2015.11.018, 2016."
**Line 643-644:** "Yang, L. M., Yu, L. E.: Measurements of oxalic acid, oxalates, malonic acid, and malonates in atmospheric particulates, Environ. Sci. Technol., 42 (24), 9268−9275, https://doi.org/10.1021/es801820z, 2008."

**L401-404: Please see my previous comment: Replacement reactions can certainly impact the aerosol phase state as you document, but in turn these are also dependent on the phase state of the aerosol particle. This aspect should be better represented in your Conclusion section.**

**Reply:** Thank you. In revised paper, the impact of phase state in chemical process has been added.

**Revision: Line 446-449:** "Our findings highlight the intricate interactions between chemical components of organic/inorganic aerosol (illustrated in Fig. 10). The interplay among organic molecular structure, molar ratio, aqueous replacement reactions, and aerosol phase state can lead to unique and potentially irreversible aerosol evolution during RH cycles. For example, the special chemical process in aerosols induced by gel state formation upon humidification."

**L397-398: "Additionally, we observed…" This is certainly a very interesting**

**finding of this study. I would like to see some more discussion, which atmospheric processes could be influenced by such "crystallization upon hydration". Also, can the authors speculate how important this process is and if they would expect it for other atmospherically relevant aerosol systems?**

**Reply:** Thank you for your comment. We very much appreciate your interest in "crystallization on hydration". Usually, the reactive uptake of gas from atmosphere occurs in liquid phase, which further provoke chemical composition evolution, and in turn change the optical property and CCN activity. This statement has been added in revised paper. We also expect similar phenomenon to happen in other dicarboxylic acids and AS mixture aerosols as long as it can potentially form gel-like structure prior to typical efflorescence during dehydration.

**Revision: Line 441-445:** "…which can reduce the possibility of further composition evolution and reactive gas uptake due to the absence of ion mobility in solid at higher RH. Hence, light absorbance and CCN activity, which strongly depend on chemical composition, will change. Future studies are needed to further investigate this phenomenon, as it is also expected to occur in other organic/inorganic aerosols that can potentially form gel-like state prior to typical efflorescence."

**L405-409: I am unclear what "targeted strategies to mitigate air pollution" the authors refer to. Please elaborate or remove this statement.**

**Reply:** According to suggestion, this statement has been deleted.

**Fig. 9: This figure is not mentioned and discussed in the text, please do that or delete the figure.**

**Reply:** Thank you. We have now referenced the figure in the relevant section in the conclusion.

**Revision: Line 446-449:** "Our findings highlight the intricate interactions between chemical components of organic/inorganic aerosol (illustrated in Fig. 10). The interplay among organic molecular structure, molar ratio, aqueous replacement reactions, and aerosol phase state can lead to unique and potentially irreversible aerosol evolution during RH cycles. For example, the special chemical process in aerosols induced by gel state formation upon humidification."

**Referee #2:**

**condense the introduction to focus more on the study's specific objectives and significance. Reduce background information that isn't directly relevant to your study's aims.**

**Reply:** Thank you for your suggestion, we rewritten the introduction to make it more concise and clearer.

**Revision: line 27-82:** please see the revised introduction in the manuscript.

**Explain the rationale for choosing specific organic salts.**

**Reply:** Thank you for the valuable suggestion. We have added justification for the selected organic-to-inorganic ratios in our introduction and materials and methods sections. In atmospheric aerosols, the organic-to-inorganic ratio and chemical structure can vary by region. Generally, the proportion of organic components is often comparable to that of inorganic components. Therefore, in our study, we used molar ratios of 2:1, 1:1, and 1:2 to represent regions with excess organic matter, a balance between organic and inorganic matter, and excess inorganic matter, respectively.

**Revision: line 76-78:** "The mole ratio of 1:2, 1:1 and 2:1 was applied to simulate the surplus organic salts, equal, and excessive inorganic slats for mixed aerosols depending on location, aerosol source and season."
**line 90-92:** "In the atmosphere, the organic to inorganic ratio varies regionally, but mostly remains at the same order of magnitude. Thus, the selected ratios herein can effectively represent different regions."

**Provide more details about the relative humidity control and measurement, such as the precision and accuracy of RH measurements.**

**Reply:** Thank for your advice. In revised manuscript, the precision and accuracy of RH measurements has been added.

**Revision: line 103:** "…with a precision of 0.5% and an accuracy of 1%..."

**It is suggested to add some summary tables comparing the phase transition behaviors of different organic salts and ammonium salts.**

**Reply:** Thank you for the excellent suggestion, and we have added the summary table to the revised paper.

**Revision:**

Table 2 The phase transition behaviors of different organic salts and ammonium salts

| Mixed aerosols | | SP mixed with | | | | | (NH$_4$)$_2$SO$_4$ mixed with | |
|---|---|---|---|---|---|---|---|---|
| | | NH$_4$Cl | | | NH$_4$NO$_3$ | (NH$_4$)$_2$SO$_4$ | SC | ST |
| | | 2:1 | 1:1 | 1:2 | (1:1) | (2:1) | 3:2 | 1:1 |
| ERH% on dehydration | SP | 61.7 | 23.7 | × | NaNO$_3$: 35.7–12.7 | Na$_2$SO$_4$: 65.7–60.1 | Na$_2$SO$_4$: 56.9 | × |
| | NH$_4$Cl | × | 42.5 | 46.6 | SP: × | SP: × | SC: × | |
| DRH% on hydration | SP | 84.3 | × | × | NaNO$_3$: 66.1–82.3 | Na$_2$SO$_4$: 83.8–90.3 | Na$_2$SO$_4$: 70.5–87.2 | Na$_2$SO$_4$: 74.8 |
| | NH$_4$Cl | × | 71.8 | 77.7 | SP: × | SP: × | SC: × | |
| ERH% on hydration | × | × | × | × | × | × | × | Na$_2$SO$_4$: 43.6 |

Note : × means no phase transition was observed

Single value represents the onset ERH or DRH

**What is the potential impact of temperature on phase transitions? What is the relationship between liquid-liquid phase separation and phase transition**

**behaviors in this study? It is suggested to discuss them in the implication.**

**Reply:** Thanks for reviewer's suggestion. The potential impact of temperature on phase transitions and the relationship between liquid-liquid phase separation and phase transition behaviors are important for climate effect. However, in our work, there is no LLPS was observed, and phase transition dependence of temperature wasn't performed. In revised paper, the effects of temperature and LLPS has been discussed based on other work.

**Revision: line 452-463:** "Except for RH, aerosol phase also depends on other atmospheric conditions, particularly temperature. Solid ammonium sulfate particles with organic coatings was observed under high relative humidity (67 to 98%) at low temperature (–2 to +4 °C), underscoring the key role of temperature in phase transitions (Kirpes et al., 2022). They proposed that solid $(NH_4)_2SO_4$ formed through contact efflorescence and that temperature induced liquid-liquid phase separation (LLPS). Previous studies have shown that temperature often has a greater impact on LLPS than on efflorescence (Schill et al., 2013). LLPS is crucial for the solidification of ammonium sulfate (Roy et al., 2020) and soot redistribution (Yuan et al., 2023) in aqueous organic-inorganic aerosol droplets. In our work, we observed various aerosol phase states, including crystalline, aqueous, and gel-like states, during dehydration and hydration cycles, and we examined the interplay among aqueous-phase reactions, aerosol hygroscopicity, and phase behavior in detail. However, LLPS was not observed prior to crystallization, either through optical microscopy or ATR-FTIR. Future studies are needed to explore the interplay between various phase transitions, including LLPS, efflorescence, and viscous state formation, considering not only RH but also temperature."

**Ensure all references are up-to-date and relevant. Check if any recent studies on similar topics have been published and could be included. E.g. recent field observation studies have highlighted the significant role of organic acids in modifying the phase behavior of ammonium sulfate aerosols, further complicating the prediction of aerosol phase states (Li et al., EST, 2021, 55, 16339; Yuan et al., 2023, ACP, 23, 9385; Kirpes et al., PNAS, 2022, 119, 14).**

**Reply:** Thanks for the reviewer's suggestion. The relevant description and references have been added.

**Revision: line 41-45:** "Recent field observations highlighted the significant role of organic acids in modifying the phase behavior of ammonium sulfate aerosols, further complicating the prediction of aerosol phase states (Li et al., 2021; Zhang et al., 2022; Kirpes et al., 2022) For atmospheric aerosols undergoing liquid–liquid phase separation (LLPS) with an organic shell and an inorganic core, balck carbon redistribution can be induced, which leads to a more compact morphology and a reduction in the absorption enhancement effect by 28 %–34% (Zhang et al. 2022).

**Revision: line 539-541:** "Li, W., Teng, X., Chen, X., Liu, L., Xu, L., Zhang, J., Wang, Y., Zhang, Y. and Shi Z.: Organic Coating Reduces Hygroscopic Growth of Phase-

Separated Aerosol Particles, Environ. Sci. Tech. 55, 16339–16346, https://doi.org/10.1021/acs.est.1c05901, 2021."

**Line 648-655:** "Yuan, Q., Wang, Y., Chen, Y., Yue, S., Zhang, J., Zhang, Y., Xu, L., Hu, W., Liu, D., Fu, P., Gao, H., Li W.: Measurement report: New insights into the mixing structures of black carbon on the eastern Tibetan Plateau – soot redistribution and fractal dimension enhancement by liquid–liquid phase separation, Atmos. Chem. Phys. 23, 9385–9399, https://doi.org/10.5194/acp-23-9385-2023, 2023.
Zhang, J., Wang, Y., Teng, X., Liu, L., Xu, Y., Ren, L., Shi, Z., Zhang, Y., Jiang, J., Liu, D., Hu, M., Shao, L., Chen, J., Martin, S. T., Zhang, X., and Li, W.: Liquid liquid phase separation reduces radiative absorption by aged black carbon aerosols, Comm. Earth Environ., 3, 128, https://doi.org/10.1038/s43247-022-00462-1, 2022.

**Line 529-532:** "Kirpes, R. M., Lei, Z., Fraund, M., Gunsch, M. J., May, N. W., Barrett, T. E., Moffett, C. E., Schauer, A. J., Alexander, B., Upchurch, L. M., China, S., Quinn, P. K., Moffet, R. C., Laskin, A., Sheesley, R. J., Pratt., K. A., Ault, A. P.: Solid organic-coated ammonium sulfate particles at high relative humidity in the summertime Arctic atmosphere, PNAS, 119, e2104496119, https://doi.org/10.1073/pnas.2104496119, 2022.

**Abstract: The abstract is informative but could be improved by including a brief mention of the methodology (ATR-FTIR spectroscopy) and the key findings related to the RH ranges.**

**Reply:** Thank you. The advice is very helpful. In the abstract, the methodology and key RH have been added.

**Line 10-11:** "We investigated carboxylate/ammonium salt mixtures using attenuated total reflection flourier transformed infrared spectroscopy (ATR-FTIR)."

**Line 15-18:** "For SP/ammonium aerosols, $NaNO_3$ and $Na_2SO_4$ crystallized from 35.7% to 12.7%, and from 65.7% to 60.1% RH, respectively, lower than pure inorganics ($62.5\pm9$–32% RH for $NaNO_3$ and $82\pm7$–$68\pm5$% RH for $Na_2SO_4$). Upon hydration, the crystalline $Na_2SO_4$ and $NaNO_3$ deliquesced at 88.8%–95.2% and $76.5\pm2$–81.9%, higher than those of pure $Na_2SO_4$ ($74\pm4$%–98% RH) and $NaNO_3$ ($65$–$77.1\pm3$% RH)."

**Line 14: A detailed introduction to "aqueous replacement reactions" for better understanding the mechanisms.**

**Reply:** Thank you very much for your suggestion. We have added a definition and examples of "aqueous replacement reactions" in the main text to help readers better understand our work.

**Revision: line 53-56:** "In aqueous aerosols, the ions between two ionic compounds can exchange to form new compounds through aqueous replacement reactions. For instance, when oxalic acid mixes with nitrates, the $H^+$ ions can combine with $NO_3^-$ ions to form $HNO_3$, which off-gases, leading to the formation of metal oxalate salts in the aerosol (Ma et al. 2019)."

**Line 16: 35.7%~12.7%, 64% and 65.5%~60.1% RH?**

**Reply:** Thank you very much for your suggestion. We have revised the description in the text to make the sentence clearer.

**Revision: line 15-18:** "For SP/ammonium aerosols, NaNO$_3$ and Na$_2$SO$_4$ crystallized from 35.7% to 12.7%, and from 65.7% to 60.1% RH, respectively, lower than pure inorganics (62.5±9–32% RH for NaNO$_3$ and 82±7–68±5% RH for Na$_2$SO$_4$). Upon hydration, the crystalline Na$_2$SO$_4$ and NaNO$_3$ deliquesced at 88.8%–95.2% and 76.5±2–81.9%, higher than those of pure Na$_2$SO$_4$ (74±4%–98% RH) and NaNO$_3$ (65–77.1±3% RH)."

**Figure 1: Enhance the figure with annotations indicating key RH ranges where phase changes occur.**

**Reply:** Thank you very much for your suggestion. We have marked the range of phase transition in the Figure 2 (original Figure 1) and added an explanation in the caption.
**Revision:** Figure 2 (original Figure 1)

[Figure]

**Line 206-208:** "Figure 2: …The yellow frame indicates the RH range when phase transition occurs (ERH of Na$_2$SO$_4$: 67.5-60.1%; onsets of ERH for SP and NH$_4$Cl: 23.7% and and 42,5%; ERH of NaNO$_3$: 35.7-12.7%)."

**Figure 7: Include a scale bar in each image to indicate the size of the observed particles. Highlight key features in the images, such as the appearance of solid entities or changes in particle morphology, with arrows or labels.**

**Reply:** Thank you very much for your suggestion. We have added a scale bar to each image and marked the RH and location where phase transitions occur.

**dehydration         hydration**

94.1%RH     70.1%RH     12.1%RH        29.0%RH     44.0%RH     80.3%RH

**Line 362-364:** "Figure 8: …The red arrow pointed the location where the phase transition occurs. The RH indicated in red marks the occurrence of phase transitions. Scale bar represents 10 micrometers."